# NEAR-OPTIMAL ALGORITHMS FOR PRIVATE ESTIMATION AND SEQUENTIAL TESTING OF COLLISION PROBABILITY

## ABSTRACT

We present new algorithms for estimating and testing *collision probability*, a fundamental measure of the spread of a discrete distribution that is widely used in many scientific fields. We describe an algorithm that satisfies $(\alpha, \beta)$-local differential privacy and estimates collision probability with error at most $\varepsilon$ using $\tilde{O}\left(\frac{\log(1/\beta)}{\alpha^2 \varepsilon^2}\right)$ samples for $\alpha \leq 1$, which improves over previous work by a factor of $\frac{1}{\alpha^2}$. We also present the first sequential testing algorithm for collision probability, which can distinguish between collision probability values that are separated by $\varepsilon$ using $\tilde{O}(\frac{1}{\varepsilon^2})$ samples, even when $\varepsilon$ is unknown. Our algorithms have nearly the optimal sample complexity and in experiments we show that they require significantly fewer samples than previous methods.

## 1 INTRODUCTION

A key property of a discrete distribution is how widely its probability mass is dispersed over its support. One of the most common measures of this dispersal is *collision probability*. Let $\mathbf{p} = (p_1, \ldots, p_k)$ be a discrete distribution. The collision probability of $\mathbf{p}$ is defined

$$C(\mathbf{p}) = \sum_{i=1}^{k} p_i^2.$$

Collision probability takes its name from the following observation. If $X$ and $X'$ are independent random variables with distribution $\mathbf{p}$ then $C(\mathbf{p}) = \Pr[X = X']$, the probability that the values of $X$ and $X'$ coincide. If a distribution is highly concentrated then its collision probability will be close to 1, while the collision probability of the uniform distribution is $1/k$.

Collision probability has played an important role in many scientific fields, although each time it is rediscovered it is typically given a different name. In ecology, collision probability is called the *Simpson index* and serves as a metric for species diversity (Simpson, 1949; Leinster, 2021). In economics, collision probability is known as the *Herfindahl–Hirschman index*, which quantifies market competition among firms (Herfindahl, 1997), and also the *Gini diversity index*, a measure of income and wealth inequality (Gini, 1912). Collision probability is also known as the *second frequency moment*, and is used in database optimization engines to estimate self join size (Cormode & Garofalakis, 2016). In statistical mechanics, collision probability is equivalent to *Tsallis entropy of second order*, which is closely related to Boltzmann–Gibbs entropy (Tsallis, 1988). The negative logarithm of collision probability is *Rényi entropy of second order*, which has many applications, including assessing the quality of random number generators (Skorski, 2017) and determining the number of reads needed to reconstruct a DNA sequence (Motahari et al., 2013). Collision probability has also been used by political scientists to determine the effective size of political parties (Laakso & Taagepera, 1979).

Collision probability is *not* equivalent to Shannon entropy, the central concept in information theory and another common measure of the spread of a distribution. However, collision probability has a much more intuitive interpretation, and is also easier to estimate. Specifically, estimating the Shannon entropy of a distribution with support size $k$ requires $\Omega\left(\frac{k}{\log k}\right)$ samples (Valiant & Valiant,

2011), while the sample complexity of estimating collision probability is independent of $k$. Additionally, the negative logarithm of the collision probability of a distribution is a lower bound on its Shannon entropy, and this lower bound becomes an equality for the uniform distribution.

## 1.1 Our Contributions

We present novel algorithms for estimating and testing the collision probability of a distribution.

**Private estimation:** We give an algorithm for estimating collision probability that satisfies $(\alpha, \beta)$-*local differential privacy*.[1] As in previous work, our algorithm is *non-interactive*, which means that there is only a single round of communication between users and a central server, and *communication-efficient*, in the sense that each user sends $O(1)$ bits to the server (in fact, just 1 bit). If $\alpha \leq 1$ then our algorithm needs $\tilde{O}\left(\frac{\log(1/\beta)}{\alpha^2 \varepsilon^2}\right)$ samples to output an estimate that has $\varepsilon$ additive error, which nearly matches the optimal sample complexity and improves on previous work by an $O\left(\frac{1}{\alpha^2}\right)$ factor (Bravo-Hermsdorff et al., 2022).

**Sequential testing:** We give an algorithm for determining whether collision probability is equal to a given value $c_0$ or differs from $c_0$ by at least $\varepsilon > 0$, assuming that one of those conditions holds. Our algorithm needs $\tilde{O}(\frac{1}{\varepsilon^2})$ samples to make a correct determination, which nearly matches the optimal sample complexity. Importantly, $\varepsilon$ is *not* known to the algorithm. In other words, the algorithm automatically adapts to easy cases by drawing fewer samples. While sequential testing algorithms have been developed for many distributional properties, such as total variation distance (Daskalakis & Kawase, 2017), as far as we know there is no existing sequential testing algorithm for collision probability. Instead, previous work has focused on the batch setting, in which the number of samples is specified in advance (Canonne, 2022a).

All of our theoretical guarantees hold with high probability, and we present numerical simulations showing that our algorithms use significantly fewer samples than existing methods. For simplicity, in the main body of this paper we state all theorems using big-$O$ notation and argue for their correctness with proof sketches only, reserving more detailed theorem statements and proofs for the Appendix.

## 2 Related Work

The collision probability of a distribution is equal to its second frequency moment, and frequency moment estimation has been widely studied in the literature on data streams, beginning with the seminal work of Alon et al. (1999). Locally differentially private estimation of frequency moments was first studied by Butucea & Issartel (2021), who gave a non-interactive mechanism for estimating any positive frequency moment. The sample complexity of their mechanism depends on the support size of the distribution, and they asked whether this dependence could be removed. Their conjecture was affirmatively resolved for collision probability by Bravo-Hermsdorff et al. (2022), but removing the dependence on support size led to a much worse dependence on the privacy parameter. It has remained an open question until now whether this trade-off is necessary.

Property and closeness testing has a rich literature (Acharya et al., 2019a; 2013; Diakonikolas et al., 2015; Goldreich & Ron, 2000; Canonne, 2022b), but the sequential setting is studied much less intensively. Existing algorithms for sequential testing almost always define closeness in terms of total variation distance, which leads to sample complexities on the order $O(\sqrt{k}/\epsilon^2)$, where $k$ is the support size of the distribution and the distribution is separated from the null hypothesis by $\epsilon$ in terms of total variation distance (Daskalakis & Kawase, 2017; Oufkir et al., 2021). By contrast, all of our results are entirely independent of $k$, making our approach more suitable when the support size is very large.

There are several batch testing approaches which are based on collision statistics. Most notably, the optimal uniform testing algorithm of Paninski (2003) distinguishes the uniform distribution from a distribution that is $\epsilon$ far from uniform in terms of total variation distance with a sample complexity $\Theta(\sqrt{k}/\epsilon^2)$. However, in the batch setting, the parameter $\epsilon$ is given to the testing algorithm as input.

---

[1] Instead of denoting the privacy parameters by $\varepsilon$ and $\delta$, as is common in the privacy literature, we will use them to denote error and probability, as is common in the statistics literature.

## 3 PRELIMINARIES

We study two problems related to learning the collision probability $C(\mathbf{p}) = \sum_i p_i^2$ of an unknown distribution $\mathbf{p} = (p_1, \ldots, p_k)$.

In the **private estimation problem**, a set of $n$ users each possess a single sample drawn independently from distribution $\mathbf{p}$. We are given an error bound $\varepsilon \geq 0$ and confidence level $\delta \in [0, 1]$. A central server must compute an estimate $\hat{C}$ that satisfies $|\hat{C} - C(\mathbf{p})| \leq \varepsilon$ with probability at least $1 - \delta$ while preserving the privacy of the users' samples. A *mechanism* is a distributed protocol between the server and the users that privately computes this estimate. The execution of a mechanism can depend on the samples, and the output of a mechanism is the entire communication transcript between the server and the users. Mechanism $M$ satisfies $(\alpha, \beta)$-*local differential privacy* if for each user $i$ and all possible samples $x_1, \ldots, x_n, x_i'$ we have

$$\Pr[M(x_1, \ldots, x_n) \in \mathcal{O}] \leq e^\alpha \Pr[M(x_1, \ldots, x_{i-1}, x_i', x_{i+1}, \ldots, x_n) \in \mathcal{O}] + \beta,$$

where $\mathcal{O}$ is any set of possible transcripts between the server and the users. In other words, if the privacy parameters $\alpha$ and $\beta$ are small then changing the sample of a single user does not significantly alter the distribution of the mechanism's output. Local differential privacy is the strongest version of differential privacy, and is suitable for a setting where the server is untrusted (Dwork et al., 2014). The *sample complexity* of the mechanism is the number of users $n$.

In the **sequential testing problem**, we are given a confidence level $\delta \in [0, 1]$ and the promise that exactly one of the following two hypotheses hold: The *null hypothesis* is that $C(\mathbf{p}) = c_0$, while the *alternative hypothesis* is that $|C(\mathbf{p}) - c_0| \geq \varepsilon > 0$. An algorithm must decide which hypothesis is correct based on samples from $\mathbf{p}$. Instead of fixing the number of samples in advance, the algorithm draws independent samples from $\mathbf{p}$ one at a time, and after observing each sample decides to either reject the null hypothesis or to continue sampling. If the null hypothesis is false then the algorithm must reject it, and if the null hypothesis is true then the algorithm must not stop sampling, and each of these events must occur with probability at least $1 - \delta$. Importantly, while $c_0$ is known to the algorithm, $\varepsilon$ is not known, and thus the algorithm must adapt to the difficulty of the problem. The *sample complexity* of the algorithm is the number of observed samples $N$ if the null hypothesis is false, a random variable.

## 4 PRIVATE ESTIMATION

In this section we describe a distributed protocol for privately estimating the collision probability of a distribution. In our protocol, a set of users each draw a sample from the distribution, and then share limited information about their samples with a central server, who computes an estimate of the collision probability while preserving the privacy of each user's sample.

As discussed in Section 1, the collision probability of a distribution is the probability that two independent samples from the distribution will coincide. Therefore the most straightforward strategy the server could employ would be to collect all the users' samples and count the number of pairs of samples containing a collision. However, this approach would not be privacy-preserving.

Instead, in Mechanism 1 below, each user applies a one-bit hash function to their private sample and shares only their hash value with the server. The server counts the number of collisions among all pairs of hash values and then applies a bias correction to form an estimate of the collision probability. To increase the robustness of this estimate, the server first partitions the hash values into groups and uses the median estimate from among the groups.

The hashing procedure in Mechanism 1 is carefully designed to both preserve user privacy and also yield an accurate estimate. On the one hand, if each user privately chose an independent hash function, then their hash values would be entirely uncorrelated and contain no useful information about the underlying distribution. On the other hand, if every user applied the same hash function to their sample, then the server could invert this function and potentially learn some user's sample. Instead, in Mechanism 1, the server sends the same hash function to all users, but each user prepends their sample with a independently chosen *salt*, or random integer, before applying the hash function. Salts are commonly used in cryptographic protocols to enhance security, and they play a similar role in our mechanism. The number of possible salts serves as a trade-off parameter between the privacy and accuracy of our mechanism, with more salts implying a stronger privacy guarantee.

---

**Mechanism 1** Private estimation for collision probability

---

**Given:** Number of users $n$, confidence level $\delta \in [0, 1]$, privacy parameters $\alpha \geq 0, \beta \in [0, 1]$.

1: Server transmits random hash function $h : \{0, 1\}^* \mapsto \{0, 1\}$ to each user.

2: Each user $i$ chooses salt $s_i$ uniformly at random from $\{1, \ldots, r\}$, where $r = 6 \left( \frac{e^\alpha + 1}{e^\alpha - 1} \right)^2 \log \frac{4}{\beta}$.

3: Each user $i$ draws sample $x_i$ from distribution $\mathbf{p}$.

4: Each user $i$ sends hash value $v_i = h(\langle s_i, x_i \rangle)$ to the server, where $\langle s_i, x_i \rangle$ is the binary encoding of $s_i$ prepended to $x_i$ and separated by a delimiter.

5: Server partitions users into $k = 8 \log \frac{1}{\delta}$ groups of size $m = \frac{n}{k}$ each.

6: Server computes the all-pairs hash value collision frequency

$$\bar{c}_g = \frac{2}{m(m-1)} \sum_{\substack{i, j \in I_g \\ i < j}} \mathbf{1}\{v_i = v_j\}$$

for each group $g$, where $I_g$ is the set of users in group $g$.

7: Server lets

$$\hat{c}_g = r(2\bar{c}_g - 1)$$

be the bias-corrected estimate for each group $g$.

8: Server outputs $\hat{C}$, the median of the $\hat{c}_g$'s.

---

The theorems in this section provide guarantees about the privacy and accuracy of Mechanism 1.

**Theorem 1.** *Mechanism 1 satisfies $(\alpha, \beta)$-local differential privacy.*

*Proof sketch.* We show that the communication transcript between the server and the users is not very likely to be different if a single user changes their sample. Note that the communication transcript consists of the random hash function chosen by the server and the users' hash values. Suppose for now that the hash function is fixed. Each user's choice of a random salt induces a distribution on their hash value, and this distribution can change if the user changes their sample. If the distribution changes too drastically then the mechanism will not be private. However, in expectation over the choice of the hash function, the distribution is always uniform, and deviations from this expectation will be small with high probability if the number of possible salts is sufficiently large. More concretely, note that the number of possible salts $r$ in Mechanism 1 increases as the privacy parameters $\alpha$ and $\beta$ decrease. Finally, since the hash function is chosen independently of the samples, the hash function reveals no information about the samples by itself. $\qquad\square$

**Theorem 2.** *If the number of samples $n$ satisfies*

$$n \geq \Omega \left( \left( \frac{e^\alpha + 1}{e^\alpha - 1} \right)^2 \frac{\log \frac{1}{\beta} \log \frac{1}{\delta}}{\varepsilon^2} \right)$$

*then the estimate $\hat{C}$ output by Mechanism 1 satisfies $|\hat{C} - C(\mathbf{p})| \leq \varepsilon$ with probability $1 - \delta$. Additionally, if $\alpha \leq 1$ then it suffices that*

$$n \geq \Omega \left( \frac{\log \frac{1}{\beta} \log \frac{1}{\delta}}{\alpha^2 \varepsilon^2} \right).$$

*Proof sketch.* The first step of the argument is to relate the likelihood of a hash collision to that of the underlying sample collision. It is not hard to see that if $x_i \neq x_j$ then $\Pr[v_i = v_j] = \frac{1}{2}$, while if $x_i = x_j$ then the $\Pr[v_i = v_j] = \frac{1}{2} + \frac{1}{r}$, because two users with the same sample and same salt are guaranteed to produce the same hash value. This discrepancy allows us to use the number of

hash collisions as an estimator of the number of sample collisions. In particular, it implies that each group estimate $\hat{c}_g$ is an unbiased estimate of $C(\mathbf{p})$.

Next we bound the variance of each $\hat{c}_g$. Clearly $\mathrm{Var}[\hat{c}_g] = O(r^2)\,\mathrm{Var}[\bar{c}_g]$. Bounding the variance $\mathrm{Var}[\bar{c}_g]$ is non-trivial, because the $v_i$'s are not independent, since they are correlated by the random choice of the hash function. By the law of total variance we have

$$\mathrm{Var}[\bar{c}_g] = \mathrm{E}[\mathrm{Var}[\bar{c}_g \mid h]] + \mathrm{Var}[\mathrm{E}[\bar{c}_g \mid h]].$$

Since the $v_i$'s are independent for a given hash function, the first term can be bounded by applying Hoeffding's theorem for U-statistics. The second term can be bounded by a fairly direct calculation.

Having shown that the $\hat{c}_g$'s are unbiased estimates of collision probability, and also having shown that each of their variances is bounded, it remains to show that their median is concentrated about their mean. This concentration follows from the analysis of the median-of-means estimator (Lugosi & Mendelson, 2019). □

### 4.1 Lower bound

The next theorem proves that the sample complexity bound in Theorem 2 is tight for small $\alpha$ up to logarithmic factors.

**Theorem 3.** *Let $\hat{C}_{\alpha,n}(\mathbf{p})$ be a collision probability estimate returned by an $(\alpha, 0)$-locally differentially private mechanism that draws $n$ samples from distribution $\mathbf{p}$. If $\alpha \leq 1$ and $n \in o\left(\frac{1}{\alpha^2 \varepsilon^2}\right)$ then there exists a distribution $\mathbf{p}$ such that*

$$\mathrm{E}\left[\left|\hat{C}_{\alpha,n}(\mathbf{p}) - C(\mathbf{p})\right|\right] \geq \varepsilon.$$

*Proof sketch.* We apply a technique due to Duchi et al. (2016) for proving minimax lower bounds for locally differentially private estimation. Their technique is a private version of Le Cam's two-point method (LeCam, 1973). It follows from Proposition 1 due to Duchi et al. (2016) that for all distributions $\mathbf{p}_0, \mathbf{p}_1$ there exists distribution $\mathbf{p}$ such that

$$\mathrm{E}\left[\left|\hat{C}_{\alpha,n}(\mathbf{p}) - C(\mathbf{p})\right|\right] \geq \frac{|C(\mathbf{p}_0) - C(\mathbf{p}_1)|}{2}\left(1 - \sqrt{2\alpha^2 n D_{\mathrm{KL}}\left(\mathbf{p}_0 \| \mathbf{p}_1\right)}\right).$$

Thus if there exist $\mathbf{p}_0$ and $\mathbf{p}_1$ such that $D_{\mathrm{KL}}\left(\mathbf{p}_0 \| \mathbf{p}_1\right) \leq O(\frac{1}{\alpha^2 n})$ and $|C(\mathbf{p}_0) - C(\mathbf{p}_1)| \geq \Omega(\frac{1}{\alpha\sqrt{n}})$ then the above lower bound is $\Omega(\frac{1}{\alpha\sqrt{n}})$, which suffices to prove the theorem. We give an explicit construction of $\mathbf{p}_0$ and $\mathbf{p}_1$ in the Appendix. Briefly, $\mathbf{p}_0$ places probability mass $\frac{1}{2}$ on one element and uniformly distributes the remaining mass on the other $k - 1$ elements, while $\mathbf{p}_1$ is nearly the same as $\mathbf{p}_0$ except for a $\Theta(\frac{1}{\alpha\sqrt{n}})$ perturbation applied to each probability. □

### 4.2 Efficient implementation

In Mechanism 1 the server computes the all-pairs hash collision frequency per group. If each group contains $m$ samples, a naive implementation would require $\Omega(m^2)$ time per group. The next theorem shows how this can be reduced to $O(m)$ time per group by computing the histogram of hash values.

**Theorem 4.** *For any values $v_1, \ldots, v_m$ if $\bar{c} = \frac{2}{m(m-1)} \sum_{i<j} \mathbf{1}\left\{v_i = v_j\right\}$ is the all-pairs collision frequency and $\hat{n}_v = \sum_i \mathbf{1}\left\{v_i = v\right\}$ is the multiplicity of value $v$ then*

$$\bar{c} = \frac{1}{m(m-1)} \sum_v \hat{n}_v^2 - \frac{1}{m-1}.$$

### 4.3 Comparison to previous work

Butucea & Issartel (2021) gave a non-interactive $(\alpha, 0)$-locally differentially private mechanism for estimating collision probability with sample complexity $\tilde{O}\left(\frac{(\log k)^2}{\varepsilon^2 \alpha^2}\right)$ and communication complexity $O(k)$. Bravo-Hermsdorff et al. (2022) gave a non-interactive mechanism with the same privacy

guarantee, sample complexity $\tilde{O}\left(\frac{1}{\alpha^4 \varepsilon^2}\right)$, and communication complexity $O(1)$.[2] Thus the latter mechanism is better suited to distributions with very large support sizes, but is a worse choice when the privacy parameter $\alpha$ is very small. Our mechanism combines the advantages of these mechanisms, at the expense of a slightly weaker privacy guarantee and an additional $\tilde{O}(\log \frac{1}{\beta})$ samples.

Notably, the earlier mechanism due to Bravo-Hermsdorff et al. (2022) is also based on counting collisions among salted hash values. But there are key differences between the mechanisms which lead to our improved sample complexity. In their mechanism, the server assigns salts to the users, each user adds noise to their hash value, and the server counts hash collisions among $\frac{n}{2}$ disjoint user pairs. In our mechanism, the salts are chosen privately, no additional noise is added to the hash values, and the server counts hash collisions among all $\binom{n}{2} = O(n^2)$ user pairs. Using all available pairs to count collisions is a more efficient use of data (although it significantly complicates the analysis, as the pairs are not all independent), and choosing the salts privately eliminates the need for additional randomness, which improves the accuracy of the estimate.

## 5 SEQUENTIAL TESTING

In this section we describe an algorithm for sequentially testing whether $C(\mathbf{p}) = c_0$ (the null hypothesis) or $|C(\mathbf{p}) - c_0| \geq \varepsilon > 0$ (the alternative hypothesis), where $c_0$ is given but $\varepsilon$ is unknown. Algorithm 2 below draws samples from the distribution $\mathbf{p}$ one at a time. Whenever the algorithm observes a sample $x_i$ it updates a running estimate of $|C(\mathbf{p}) - c_0|$ based on the all-pairs collision frequency observed so far. The algorithm compares this estimate to a threshold that shrinks like $\Theta\left(\sqrt{i^{-1} \log \log i}\right)$ and rejects the null hypothesis as soon as the threshold is exceeded. Although our algorithm is simple to describe, its proof of correctness is non-trivial, as it involves showing that a sequence of dependent random variables (the running estimates) become concentrated. Our proof uses a novel decoupling technique to construct martingales based on the running estimates.

---

**Algorithm 2** Sequential testing of collision probability

---

**Given:** Null hypothesis value $c_0$, confidence level $\delta \in [0, 1]$.

1: **for** $i = 1, 2, 3, \ldots$ **do**
2:     Draw sample $x_i$ from distribution $\mathbf{p}$.
3:     Let $T_i = \sum_{j=1}^{i-1} \mathbf{1}\{x_i = x_j\} - 2(i-1)c_0$.
4:     **if** $\left|\frac{2}{i(i-1)} \sum_{j=1}^{i} T_j\right| > 3.2 \sqrt{\frac{\log \log i + 0.72 \log(20.8/\delta)}{i}}$ **then**
5:         Reject the null hypothesis.
6:     **end if**
7: **end for**

---

The next theorem provides a guarantee about the accuracy of Algorithm 2.

**Theorem 5.** *If $C(\mathbf{p}) = c_0$ then Algorithm 2 does not reject the null hypothesis with probability $1 - \delta$. If $|C(\mathbf{p}) - c_0| \geq \varepsilon$ then Algorithm 2 rejects the null hypothesis after observing $N$ samples, where*

$$N \in O\left(\frac{1}{\varepsilon^2} \log \log \frac{1}{\varepsilon} \log \frac{1}{\delta}\right)$$

*with probability $1 - \delta$.*

The $\log \log \frac{1}{\varepsilon}$ factor in Theorem 5 results from our application of a confidence interval due to Howard et al. (2021) that shrinks like $\Theta\left(\sqrt{i^{-1} \log \log i}\right)$. Note that $\log \log \frac{1}{\varepsilon} < 4$ if $\varepsilon \geq 10^{-10}$, so this factor is negligible for nearly all problem instances of practical interest.

---

[2]Note that Bravo-Hermsdorff et al.'s original NeurIPS paper claimed $\tilde{O}\left(\frac{1}{\alpha^2 \varepsilon^2}\right)$ sample complexity, but a more recent version on Arxiv claims $\tilde{O}\left(\frac{1}{\alpha^4 \varepsilon^2}\right)$ sample complexity and explains that the original version contained mistakes. See References for a link to the Arxiv version.

*Proof sketch of Theorem 5.* First note that $T_1, T_2, , \ldots$ which are used in Line 3 of Algorithm 2 are dependent sequences, so $T_i$ depends on all $x_1, \ldots, x_i$, which prevent us from computing a concentration bound for it. Therefore we shall apply a decoupling technique to derive a martingale sequence. Let us define $\bar{U}_m := \bar{U}(X_1, \ldots, X_m) = \sum_{i<j} g(X_i, X_j)$ with

$$g(X_i, X_j) = \mathbf{1}\{X_i = X_j\} - \mathrm{E}\left[\mathbf{1}\{X_i = X_j\} | X_i\right] - \mathrm{E}\left[\mathbf{1}\{X_i = X_j\} | X_j\right] + \mathrm{E}\left[\mathbf{1}\{X_i = X_j\}\right]$$
$$= \mathbf{1}\{X_i = X_j\} - \Pr(X_i = X_j | X_i) - \Pr(X_i = X_j | X_j) + c_0 \ .$$

This decoupling technique is motivated by Theorem 8.1.1 of Tsybakov (2008) since the kernel function $g$ has became centered and degenerated, i.e. $\mathrm{E}\left[g(X_i, X_j) | X_j\right] = \mathrm{E}\left[g(X_i, X_j) | X_i\right] = 0$ which implies that $\bar{U}_n$ is a zero-mean martingale with $m \geq 2$. The empirical sequence is $\bar{u}_m = \sum_{i=1}^{m} y_m$ with

$$y_j = \sum_{i=1}^{m-1} \mathbf{1}\{x_i = x_j\} - \sum_{i=1}^{m-1} p_{x_i} - (m-1)p_{x_j} + (m-1)c_0$$

which is has bounded differences such that $|\bar{U}_k - \bar{U}_{k-1}| = |Y_k| \leq 4m$ and $y_1 = 0$. However we cannot compute this empirical sequence, since the parameters of distribution are not known. As a remedy, we further decompose $\bar{U}_n$ as the sum of two sequences based on the observation that

$$\mathbb{E}[p_{X_i}] = \sum_i p_{x_i}^2 = c_0$$

which implies that $\sum_{i=1}^{m}(p_{X_i} - c_0)$ is again a zero-mean martingale sequence with the same filtration $\mathcal{F}_m$ such that the difference $|p_{X_i} - c_0| < 1$ for all $i$. This motivates the following decomposition of $\bar{U}_n$ as

$$Y_j = \underbrace{\sum_{i=1}^{j-1} \mathbf{1}\{X_i = X_j\} - 2(j-1)c_0}_{T_j} + \underbrace{2(j-1)c_0 - \sum_{i=1}^{j-1} p_{X_i} - (j-1)p_{X_j}}_{E_j}$$

Note that $T_m$, used in Algorithm 2, can be computed and it is a zero-mean martingale sequence up to an error term $E_n$ which cannot be computed, since the parameters of the underlying distribution $\mathbf{p}$ is not available. However $E_n$ can be again decomposed into sequence of sums of zero mean-mean terms which we can upper bound with high probability. Important to note that if $H_0 : c_0 = 1/K$, the error term is equal to zero in any time step, i.e. $E_n = 0, \forall n \in [1, 2, \ldots)$, therefore $T_m$ is a zero-mean martingale itself. Finally, we rely on the work of Howard et al. (2021) in which a sequence of confidence intervals is introduced for martingales that hold uniformly for each time step, even with random stopping time. □

We remark that our proof technique bears some superficial resemblance to the approach used in recent work by Oufkir et al. (2021). They make use of the fact that for any random variable $T$ taking values from $\mathbb{N}$ and for all $T \in \mathbb{N}_+$, it holds that $\mathbb{E}[T] \leq N + \sum_{t \geq N} \mathbb{P}(T \geq t)$. Then with a carefully selected $N$ and Chernoff bounds with infinite many applications of union bound implies upper bound on the expected sample complexity. By contrast, we construct a test martingale that is specific to collision probability and apply an anytime or time-uniform concentration bound to the martingale introduced by Waudby-Smith & Ramdas (2020).

## 5.1 Lower bound

The next theorem proves that sample complexity bound in Theorem 5 is tight up to log-log factors.

**Theorem 6.** *Let $N$ be the number of samples observed by a sequential testing algorithm for collision probability. For all $\varepsilon, \delta \in [0, 1]$ there exists a distribution $\mathbf{p}$ and $c_0 \in [0, 1]$ such that $|C(\mathbf{p}) - c_0| \geq \varepsilon$ and if the algorithm rejects the null hypothesis with probability $1 - \delta$ then*

$$\mathrm{E}[N] \geq \Omega\left(\frac{\log(1/\delta)}{\varepsilon^2}\right).$$

*Proof sketch.* Our proof is based on a reduction to the problem of *identity testing* and a lower bound for that problem due to Oufkir et al. (2021). In an identity testing problem we are given a distribution $\mathbf{p}_0$ and sample access to a distribution $\mathbf{p}_1$ and the goal is to decide whether $\mathbf{p}_0 = \mathbf{p}_1$

or $\|\mathbf{p}_0 - \mathbf{p}_1\|_1 \geq \varepsilon > 0$. Oufkir et al. (2021) proved that if $\|\mathbf{p}_0 - \mathbf{p}_1\|_1 \geq \varepsilon$ then the number of samples $N$ needed to make a correct decision must satisfy $\mathrm{E}[N] \geq \frac{\log(1/(3\delta))}{D_{\mathrm{KL}}(\mathbf{p}_0\|\mathbf{p}_1)}$. We complete the proof by showing that there exist distributions $\mathbf{p}_0$ and $\mathbf{p}_1$ such that $\|\mathbf{p}_0 - \mathbf{p}_1\|_1 \geq \Omega(\varepsilon)$, $|C(\mathbf{p}_0) - C(\mathbf{p}_1)| \geq \Omega(\varepsilon)$ and $D_{\mathrm{KL}}(\mathbf{p}_0\|\mathbf{p}_1) \leq O(\varepsilon^2)$. An explicit construction of $\mathbf{p}_0$ and $\mathbf{p}_1$ is in the Appendix, and they are the same distributions as in the proof of Theorem 3. $\square$

## 6 EXPERIMENTS

We compare our mechanism for private collision probability estimation (Mechanism 1) to the recently proposed mechanism from Bravo-Hermsdorff et al. (2022). As discussed in Section 4.3, we expect Mechanism 1 to outperform their mechanism when the support size of the distribution is large and the privacy requirement is strict. We also compare to an indirect method: Privately estimate the distribution itself, and then compute the collision probability of the estimated distribution. In our experiments we use an open-source implementation of a private heavy hitters algorithm due to Cormode et al. (2021).[3]

In Figure 1 we use each mechanism to privately estimate the collision probability of two distributions supported on 1000 elements: the uniform distribution ($p_i = 1/k$) and the power law distribution ($p_i \propto 1/i$). Our simulations show that Mechanism 1 has significantly lower error for small values of the privacy parameters $\alpha$ and $\beta$.

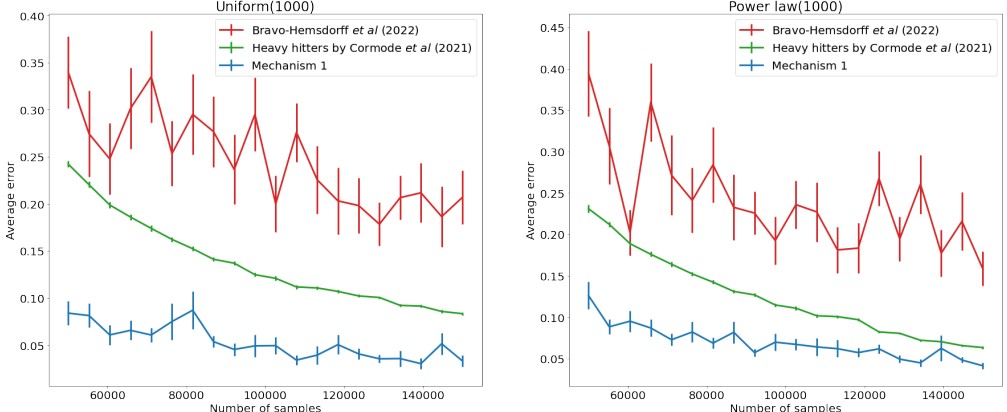

Figure 1: Sample complexity of private collision probability estimation mechanisms for $\alpha = 0.25$. Both mechanisms use the MD5 hash function and confidence level $\delta = 0.1$. For Mechanism 1 we let $\beta = 10^{-5}$. Error bars are one standard error.

We next evaluate our sequential testing algorithm (Algorithm 2). Since we are not aware of any existing algorithm for sequential testing of collision probability, we compare Algorithm 2 to two batch testing algorithms, both of which are described in a survey by Canonne (2022a):

- **Plug-in:** Form empirical distribution $\hat{\mathbf{p}}$ from samples $x_1, \ldots, x_n$ and let $\hat{C} = C(\hat{\mathbf{p}})$.
- **U-statistics:** Let $\hat{C} = \frac{2}{n(n-1)} \sum_{i<j} \mathbf{1}\{x_i = x_j\}$ be the all-pairs collision frequency.

Each batch testing algorithm takes as input both the null hypothesis value $c_0$ and a tolerance parameter $\varepsilon$, and compares $|\hat{C} - c_0|$ to $\varepsilon$ to decide whether to reject the null hypothesis $C(\mathbf{p}) = c_0$. The sample complexity of a batch testing algorithm is determined via worst-case theoretical analysis in terms of $\varepsilon$ (see Appendix). On the other hand, sequential testing algorithms automatically adapt their sample complexity to the difference $|C(\mathbf{p}) - c_0|$.

In Figure 2 we evaluate batch and sequential testing algorithms on both on the uniform distribution and power law distributions. We use 20 different support sizes for each distribution, evenly spaced on a log scale between 10 and $10^6$ inclusively. Varying the support size also varies $|C(\mathbf{p}) - c_0|$.

---

[3]https://github.com/Samuel-Maddock/pure-LDP

As expected, when $|C(\mathbf{p}) - c_0|$ is large, our sequential testing algorithm requires many fewer samples than the batch algorithm to reject the null hypothesis, and as $|C(\mathbf{p}) - c_0|$ shrinks the number of samples required sharply increases (see grey areas in Figure 2). In all cases our sequential testing algorithm is never outperformed by the batch testing algorithms.

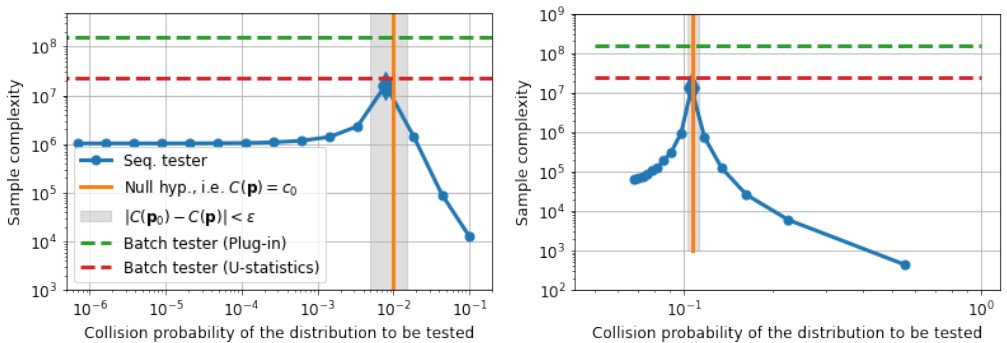

Figure 2: Sample complexity of the sequential tester compared to the sample complexity of the batch testers. For the batch testers, the tolerance parameter $\epsilon$ is set to $0.01$.

Note that in Figure 3 the plug-in tester has a worse sample complexity than the U-statistics tester. Since these sample complexities are determined by theoretical analysis, we experimentally confirmed that this discrepancy is not simply an artifact of the analysis. In Figure 3 we run simulations comparing the algorithms in terms of their error $|\hat{C} - C(\mathbf{p})|$, and find that the plug-in tester is also empirically worse than the U-statistics tester.

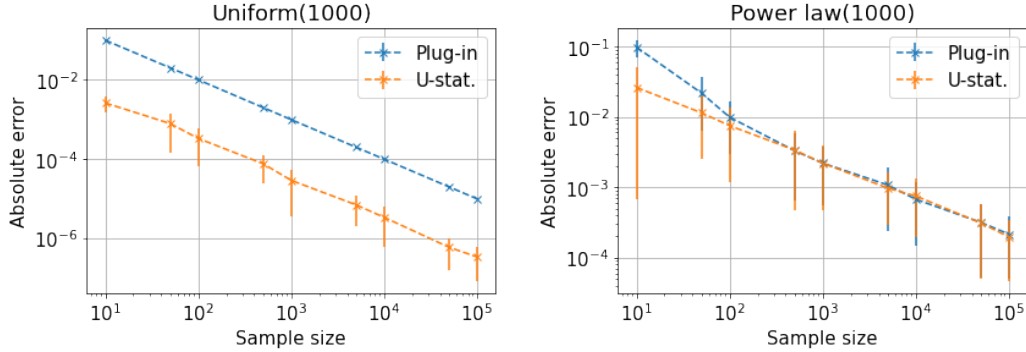

Figure 3: Empirical absolute error of plug-in and U-statistic estimators when the data is generated from uniform distribution and power law with domain size 1000.

## 7    CONCLUSIONS AND FUTURE WORK

We introduced a locally differentially private estimator for collision probability that is near-optimal in a minimax sense and empirically superior to the state-of-the-art method introduced by Bravo-Hermsdorff et al. (2022). Our method is based on directly estimating the collision probability using all pairs of observed samples, unlike in previous work. We also introduced a near-optimal sequential testing algorithm that is likewise based on directly estimating the collision probability, and requires far fewer samples than the minimax optimal batch testing algorithm for many problem instances. In the future, we plan to combine these methods and develop a locally differentially private sequential testing algorithm which, to our best knowledge, does not currently exist. Also, we plan to develop an adaptive testing algorithm which accounts for the variance of the estimator, which may allow us to achieve even lower sample complexity (such as $O(1/\epsilon)$) for particularly easy problem instances.

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

## A APPENDIX

## B PROOF OF THEOREM 1

Let $X_i$ and $V_i$ be the private value and hash value, respectively, for user $i$, and let $H$ be the random hash function chosen by the server. Let $\mathbf{X} = (X_1, \ldots, X_n)$ and $\mathbf{V} = (V_1, \ldots, V_n)$.

Recall that in the local model of differential privacy, the output of a mechanism is the entire communication transcript between the users and the server, which in the case of Mechanism 1 consists of the hash function and all the hash values. Therefore our goal is to prove that for any subset $\mathcal{O}$ of possible values for $(\mathbf{V}, H)$ we have

$$\Pr[(\mathbf{V}, H) \in \mathcal{O} \mid \mathbf{X} = \mathbf{x}] \leq e^{\alpha} \Pr[(\mathbf{V}, H) \in \mathcal{O} \mid \mathbf{X} = \mathbf{x}'] + \beta. \tag{1}$$

where $\mathbf{x}$ and $\mathbf{x}'$ differ in one element.

The proof of Eq. (1) will need a couple of observations. First, given the hash function and private values, the hash values are mutually independent:

$$\Pr[\mathbf{V} = \mathbf{v} \mid H = h \wedge \mathbf{X} = \mathbf{x}] = \prod_i \Pr[V_i = v_i \mid H = h \wedge X_i = x_i]. \tag{2}$$

Second, the hash function is independent of the private values:

$$\Pr[H = h \mid \mathbf{X} = \mathbf{x}] = \Pr[H = h \mid \mathbf{X} = \mathbf{x}']. \tag{3}$$

We also need a definition. Suppose $\mathbf{x}$ and $\mathbf{x}'$ differ in the $j$th element. Define $H_\alpha$ to be the set of all hash functions such that if $h \in H_\alpha$ then

$$\Pr[V_j = v \mid H = h \wedge X_j = x_j] \leq e^{\alpha} \Pr[V_j = v \mid H = h \wedge X_j = x_j'] \tag{4}$$

for all $v$. Note that the definition of $H_\alpha$ depends implicitly on $x_j$ and $x_j'$, but does not depend on $X_j$.

Combining the above we have

$$\begin{aligned}
&\Pr[\mathbf{V} = \mathbf{v} \wedge H = h \mid \mathbf{X} = \mathbf{x} \wedge H \in \mathcal{H}_\alpha] \\
&= \Pr[\mathbf{V} = \mathbf{v} \mid H = h \wedge \mathbf{X} = \mathbf{x} \wedge H \in \mathcal{H}_\alpha] \cdot \Pr[H = h \mid \mathbf{X} = \mathbf{x} \wedge H \in \mathcal{H}_\alpha] \\
&= \prod_i \Pr[V_i = v_i \mid H = h \wedge X_i = x_i \wedge H \in \mathcal{H}_\alpha] \cdot \Pr[H = h \mid \mathbf{X} = \mathbf{x} \wedge H \in \mathcal{H}_\alpha] && \because \text{Eq. (2)} \\
&= \prod_i \Pr[V_i = v_i \mid H = h \wedge X_i = x_i \wedge H \in \mathcal{H}_\alpha] \cdot \Pr[H = h \mid \mathbf{X} = \mathbf{x}' \wedge H \in \mathcal{H}_\alpha] && \because \text{Eq. (3)} \\
&\leq e^{\alpha} \prod_i \Pr[V_i = v_i \mid H = h \wedge X_i = x_i' \wedge H \in \mathcal{H}_\alpha] \cdot \Pr[H = h \mid \mathbf{X} = \mathbf{x}' \wedge H \in \mathcal{H}_\alpha] && \because \text{Eq. (4)} \\
&= e^{\alpha} \Pr[\mathbf{V} = \mathbf{v} \mid H = h \wedge \mathbf{X} = \mathbf{x}' \wedge H \in \mathcal{H}_\alpha] \cdot \Pr[H = h \mid \mathbf{X} = \mathbf{x}' \wedge H \in \mathcal{H}_\alpha] && \because \text{Eq. (2)} \\
&= e^{\alpha} \Pr[\mathbf{V} = \mathbf{v} \wedge H = h \mid \mathbf{X} = \mathbf{x}' \wedge H \in \mathcal{H}_\alpha].
\end{aligned}$$

Summing both sides over $\mathcal{O}$ yields

$$\Pr[(\mathbf{V}, H) \in \mathcal{O} \mid \mathbf{X} = \mathbf{x} \wedge H \in \mathcal{H}_\alpha] \leq e^{\alpha} \Pr[(\mathbf{V}, H) \in \mathcal{O} \mid \mathbf{X} = \mathbf{x}' \wedge H \in \mathcal{H}_\alpha]. \tag{5}$$

Therefore we have

$$\begin{aligned}
\Pr[(\mathbf{V}, H) \in \mathcal{O} \mid \mathbf{X} = \mathbf{x}] &= \Pr[(\mathbf{V}, H) \in \mathcal{O} \mid \mathbf{X} = \mathbf{x} \wedge H \in \mathcal{H}_\alpha] \cdot \Pr[H \in \mathcal{H}_\alpha \mid \mathbf{X} = \mathbf{x}] \\
&\quad + \Pr[(\mathbf{V}, H) \in \mathcal{O} \mid \mathbf{X} = \mathbf{x} \wedge H \notin \mathcal{H}_\alpha] \cdot \Pr[H \notin \mathcal{H}_\alpha \mid \mathbf{X} = \mathbf{x}] \\
&\leq \Pr[(\mathbf{V}, H) \in \mathcal{O} \mid \mathbf{X} = \mathbf{x} \wedge H \in \mathcal{H}_\alpha] \cdot \Pr[H \in \mathcal{H}_\alpha \mid \mathbf{X} = \mathbf{x}] \\
&\quad + \Pr[H \notin \mathcal{H}_\alpha \mid \mathbf{X} = \mathbf{x}] \\
&= \Pr[(\mathbf{V}, H) \in \mathcal{O} \mid \mathbf{X} = \mathbf{x} \wedge H \in \mathcal{H}_\alpha] \cdot \Pr[H \in \mathcal{H}_\alpha \mid \mathbf{X} = \mathbf{x}'] && \because \text{Eq. (3)} \\
&\quad + \Pr[H \notin \mathcal{H}_\alpha \mid \mathbf{X} = \mathbf{x}] \\
&\leq e^{\alpha} \Pr[(\mathbf{V}, H) \in \mathcal{O} \mid \mathbf{X} = \mathbf{x}' \wedge H \in \mathcal{H}_\alpha] \cdot \Pr[H \in \mathcal{H}_\alpha \mid \mathbf{X} = \mathbf{x}'] && \because \text{Eq. (5)}
\end{aligned}$$

$$+ \Pr[H \notin \mathcal{H}_\alpha \mid \mathbf{X} = \mathbf{x}]$$
$$= e^\alpha \Pr[(\mathbf{V}, H) \in \mathcal{O} \wedge H \in \mathcal{H}_\alpha \mid \mathbf{X} = \mathbf{x}'] + \Pr[H \notin \mathcal{H}_\alpha \mid \mathbf{X} = \mathbf{x}]$$
$$\leq e^\alpha \Pr[(\mathbf{V}, H) \in O \mid \mathbf{X} = \mathbf{x}'] + \Pr[H \notin \mathcal{H}_\alpha \mid \mathbf{X} = \mathbf{x}]$$
$$= e^\alpha \Pr[(\mathbf{V}, H) \in O \mid \mathbf{X} = \mathbf{x}'] + \Pr[H \notin \mathcal{H}_\alpha] \qquad \because \text{Eq. (3)}$$

It remains to show that $\Pr[H \notin \mathcal{H}_\alpha] \leq \beta$. Recall that $x_j \neq x_j'$ are the only different values in $\mathbf{x}$ and $\mathbf{x}'$. For any hash value $v$ and salt $s$ define the random variables

$$p_{s,v}(H) = \mathbf{1}\left\{H(\langle s, x_j \rangle) = v\right\}$$
$$p_{s,v}'(H) = \mathbf{1}\left\{H(\langle s, x_j' \rangle) = v\right\}$$

Also define $\bar{p}_v(H) = \frac{1}{r} \sum_s p_{s,v}(H)$ and $\bar{p}_v'(H) = \frac{1}{r} \sum_s p_{s,v}'(H)$. Observe that

$$\Pr[V_j = v \mid H = h \wedge X_j = x_j] = \bar{p}_v(h)$$
$$\Pr[V_j = v \mid H = h \wedge X_j = x_j'] = \bar{p}_v'(h)$$

since each user chooses their salt uniformly at random from $\{1, \ldots, r\}$. Therefore

$$\Pr[H \notin \mathcal{H}_\alpha] = \Pr\left[\exists v : \frac{\bar{p}_v(H)}{\bar{p}_v'(H)} > e^\alpha\right]. \qquad (6)$$

We will analyze the right-hand side of Eq. (6). Clearly $\mathrm{E}[p_{s,v}(H)] = \mathrm{E}[p_{s,v}'(H)] = \frac{1}{b}$, since each $H(\langle s, x \rangle)$ is chosen uniformly at random from $\{1, \ldots, b\}$. Also $\bar{p}_v(H)$ and $\bar{p}_v'(H)$ are each the average of $r$ independent Boolean random variables, since each $H(\langle s, x \rangle)$ is chosen independently. Therefore, by the Chernoff bound, for all $\varepsilon \in [0, 1]$ and any hash value $v$

$$\Pr\left[\bar{p}_v(H) \geq (1 + \varepsilon)\frac{1}{b}\right] \leq \exp\left(-\frac{\varepsilon^2 r}{3b}\right)$$
$$\Pr\left[\bar{p}_v'(H) \leq (1 - \varepsilon)\frac{1}{b}\right] \leq \exp\left(-\frac{\varepsilon^2 r}{3b}\right)$$

Fix $\varepsilon = \frac{e^\alpha - 1}{e^\alpha + 1}$. Observe that if $\bar{p}_v(H) \leq (1 + \varepsilon)\frac{1}{b}$ and $\bar{p}_v'(H) \geq (1 - \varepsilon)\frac{1}{b}$ then

$$\frac{\bar{p}_v(H)}{\bar{p}_v'(H)} \leq \frac{1 + \varepsilon}{1 - \varepsilon} = e^\alpha. \qquad (7)$$

Continuing from Eq. (6)

$$\Pr\left[\exists v : \frac{\bar{p}_v(H)}{\bar{p}_v'(H)} > e^\alpha\right] \leq \sum_v \Pr\left[\frac{\bar{p}_v(H)}{\bar{p}_v'(H)} > e^\alpha\right]$$
$$\leq \sum_v \Pr\left[\bar{p}_v(H) \geq (1 + \varepsilon)\frac{1}{b} \vee \bar{p}_v'(H) \leq (1 - \varepsilon)\frac{1}{b}\right] \qquad \because \text{Eq. (7)}$$
$$\leq \sum_v 2 \exp\left(-\frac{\varepsilon^2 r}{3b}\right) \qquad \because \text{Chernoff bound}$$
$$= 2b \exp\left(-\frac{\varepsilon^2 r}{3b}\right)$$
$$\leq \beta$$

where the last line follows from $r \geq \frac{3b}{\varepsilon^2} \log \frac{2b}{\beta}$.

## C  PROOF OF THEOREM 2

**Definition 1.** *For any $k \geq 0$ the $k$th frequency moment of discrete random variable $X$ is*

$$F_k(X) = \sum_x \Pr[X = x]^k.$$

The next lemma says that when $k$ is a positive integer then $F_k(X)$ is the probability that $k$ independent copies of $X$ all have the same value.

**Lemma 1.** *If $k$ is a positive integer then*

$$F_k(X) = \Pr[X_1 = \cdots = X_k]$$

*where $X_1, \ldots, X_k$ are independent and have the same distribution as $X$.*

*Proof.*

$$
\begin{aligned}
F_k(X) &= \sum_x \Pr[X = x]^k \\
&= \sum_x \Pr[X_1 = x] \cdots \Pr[X_k = x] && \because \text{Identical distributions} \\
&= \sum_x \Pr[X_1 = x \wedge \cdots \wedge X_k = x] && \because \text{Independence} \\
&= \Pr[X_1 = \cdots = X_k]
\end{aligned}
$$

$\square$

Next we compute the first and second frequency moments of the hash value returned by each user in Mechanism 1.

**Lemma 2.** *Let $S$ be a uniform random variable in $\{1, \ldots, r\}$. Let $X$ be independent from $S$. Let $H : \{0, 1\}^* \mapsto \{1, \ldots, b\}$ be a random hash function. If*

$$V = H(\langle S, X \rangle)$$

*then*

$$F_2(V) = \frac{1}{b} + F_2(X)\left(\frac{b-1}{br}\right)$$

$$F_3(V) = \frac{1}{b^2} + 3F_2(X)\left(\frac{b-1}{b^2 r}\right) + F_3(X)\left(\frac{(b-1)(b-2)}{b^2 r^2}\right)$$

$$F_3(V) = \frac{1 - 3F_2(X) + 2F_3(X)}{b^2} + F_3(X)\left[\frac{1}{r^2} + 3\frac{r-1}{r^2 b} + \frac{1 - 3/r + 2/r^2}{b^2}\right] + 3(F_2(X) - F_3(X))\left[\frac{1}{rb} + \frac{r-1}{rb^2}\right]$$

$$= \frac{1}{b^2} + 3F_2(X)\left[\frac{1}{rb} + \frac{(r-1)}{rb^2} - \frac{1}{b^2}\right] + F_3(X)\left[\frac{1}{r^2} + 3\frac{r-1}{r^2 b} + \frac{2}{r^2 b^2} - \frac{3}{rb} - \frac{1}{b^2}\right]$$

*Proof.* Let $S_1, S_2$ and $S_3$ be independent random variables with the same distribution as $S$. Let $X_1, X_2$ and $X_3$ be independent random variables with the same distribution as $X$. Let $V_i = H(\langle S_i, X_i \rangle)$ for $i \in \{1, 2, 3\}$. By Lemma 1 we have

$$
\begin{aligned}
F_2(V) =\ & \Pr[V_1 = V_2] \\
=\ & \Pr[V_1 = V_2 \mid S_1 = S_2 \wedge X_1 = X_2] \cdot \Pr[S_1 = S_2] \cdot \Pr[X_1 = X_2] + \\
& \Pr[V_1 = V_2 \mid S_1 \neq S_2 \wedge X_1 = X_2] \cdot \Pr[S_1 \neq S_2] \cdot \Pr[X_1 = X_2] + \\
& \Pr[V_1 = V_2 \mid S_1 = S_2 \wedge X_1 \neq X_2] \cdot \Pr[S_1 = S_2] \cdot \Pr[X_1 \neq X_2] + \\
& \Pr[V_1 = V_2 \mid S_1 \neq S_2 \wedge X_1 \neq X_2] \cdot \Pr[S_1 \neq S_2] \cdot \Pr[X_1 \neq X_2] \\
=\ & 1 \cdot \frac{1}{r} \cdot F_2(X) + \\
& \frac{1}{b} \cdot \left(1 - \frac{1}{r}\right) \cdot F_2(X) + \\
& \frac{1}{b} \cdot \frac{1}{r} \cdot (1 - F_2(X)) + \\
& \frac{1}{b} \cdot \left(1 - \frac{1}{r}\right) \cdot (1 - F_2(X))
\end{aligned}
$$

$$= \frac{1}{b} + F_2(X) \left( \frac{1}{r} - \frac{1}{br} \right)$$

$$= \frac{1}{b} + F_2(X) \left( \frac{b-1}{br} \right)$$

which proves the first part of the lemma. For the second part, let $\mathbf{S} = (S_1, S_2, S_3)$, $\mathbf{X} = (X_1, X_2, X_3)$ and $\mathbf{V} = (V_1, V_2, V_3)$. Also for any vectors $\mathbf{Z}^{(1)}, \dots, \mathbf{Z}^{(k)}$ let

$$m(\mathbf{Z}^{(1)}, \dots, \mathbf{Z}^{(k)}) = \left| \left\{ (i,j) : i < j \wedge Z_i^{(1)} = Z_j^{(1)} \wedge \cdots \wedge Z_i^{(k)} = Z_j^{(k)} \right\} \right|.$$

For example, if $m(\mathbf{S}, \mathbf{X}) = 1$ then exactly one pair of the variables $S_1, S_2$ and $S_3$ are equal, as are one pair of the variables $X_1, X_2$ and $X_3$, and moreover the equal pairs have the same indices. The distribution of $m(\mathbf{X})$ is

$\Pr[m(\mathbf{X}) = 3] = \Pr[X_1 = X_2 = X_3] = F_3(X)$

$\Pr[m(\mathbf{X}) = 2] = 3\Pr[X_1 = X_2 \wedge X_2 = X_3 \wedge X_1 \neq X_3] = 0$

$\Pr[m(\mathbf{X}) = 1] = 3\Pr[X_1 = X_2 \neq X_3] = 3(\Pr[X_1 = X_2] - \Pr[X_1 = X_2 = X_3]) = 3(F_2(X) - F_3(X))$

$\Pr[m(\mathbf{X}) = 0] = 1 - \sum_{k=1}^{3} \Pr[m(\mathbf{X}) = k] = 1 - 3F_2(X) + 2F_3(X).$

Thus $m(\mathbf{X}) \in \{0, 1, 3\}$ with probability 1. Clearly $m(\mathbf{S}, \mathbf{X}) \leq \min\{m(\mathbf{S}), m(\mathbf{X})\}$. So the conditional distribution of $m(\mathbf{S}, \mathbf{X})$ given $m(\mathbf{X})$ is

$$\Pr[m(\mathbf{S}, \mathbf{X}) = 3 \mid m(\mathbf{X}) = 3] = \Pr[S_1 = S_2 = S_3] = \frac{1}{r^2}$$

$$\Pr[m(\mathbf{S}, \mathbf{X}) = 2 \mid m(\mathbf{X}) = 3] = 3\Pr[S_1 = S_2 \wedge S_2 = S_3 \wedge S_1 \neq S_3] = 0$$

$$\Pr[m(\mathbf{S}, \mathbf{X}) = 1 \mid m(\mathbf{X}) = 3] = 3\Pr[S_1 = S_2 \neq S_3] = \frac{3}{r} \left( 1 - \frac{1}{r} \right)$$

$$\Pr[m(\mathbf{S}, \mathbf{X}) = 0 \mid m(\mathbf{X}) = 3] = 1 - \sum_{k=1}^{3} \Pr[m(\mathbf{S}, \mathbf{X}) = k \mid m(\mathbf{X}) = 3] = 1 - \frac{3}{r} + \frac{2}{r^2}$$

$$\Pr[m(\mathbf{S}, \mathbf{X}) = 1 \mid m(\mathbf{X}) = 1] = \Pr[S_1 = S_2] = \frac{1}{r}$$

$$\Pr[m(\mathbf{S}, \mathbf{X}) = 0 \mid m(\mathbf{X}) = 1] = \Pr[S_1 \neq S_2] = 1 - \frac{1}{r}$$

$$\Pr[m(\mathbf{S}, \mathbf{X}) = 0 \mid m(\mathbf{X}) = 0] = 1$$

Thus $m(\mathbf{S}, \mathbf{X}) \in \{0, 1, 3\}$ with probability 1. So the conditional distribution of $m(\mathbf{V})$ given $m(\mathbf{S}, \mathbf{X})$ is

$$\Pr[m(\mathbf{V}) = 3 \mid m(\mathbf{S}, \mathbf{X}) = 3] = 1$$

$$\Pr[m(\mathbf{V}) = 3 \mid m(\mathbf{S}, \mathbf{X}) = 1] = \frac{1}{b}$$

$$\Pr[m(\mathbf{V}) = 3 \mid m(\mathbf{S}, \mathbf{X}) = 0] = \frac{1}{b^2}.$$

Putting it all together we have

$$F_3(V) = \Pr[V_1 = V_2 = V_3]$$

$$= \Pr[m(\mathbf{V}) = 3]$$

$$= \sum_{k_1=0}^{3} \sum_{k_2=0}^{k_1} \Pr[m(\mathbf{V}) = 3 \mid m(\mathbf{X}) = k_1 \wedge m(\mathbf{S}, \mathbf{X}) = k_2] \cdot \Pr[m(\mathbf{X}) = k_1 \wedge m(\mathbf{S}, \mathbf{X}) = k_2]$$

$$= \sum_{k_1=0}^{3} \sum_{k_2=0}^{k_1} \Pr[m(\mathbf{V}) = 3 \mid m(\mathbf{S}, \mathbf{X}) = k_2] \cdot \Pr[m(\mathbf{S}, \mathbf{X}) = k_2 \mid m(\mathbf{X}) = k_1] \cdot \Pr[m(\mathbf{X}) = k_1]$$

$$= \frac{1}{b^2} \cdot 1 \cdot (1 - 3F_2(X) + 2F_3(X)) +$$

$$\frac{1}{b^2} \cdot \left(1 - \frac{1}{r}\right) \cdot 3(F_2(X) - F_3(X)) + \frac{1}{b} \cdot \frac{1}{r} \cdot 3(F_2(X) - F_3(X)) +$$

$$\frac{1}{b^2} \cdot \left(1 - \frac{3}{r} + \frac{2}{r^2}\right) \cdot F_3(X) + \frac{1}{b} \cdot \frac{3}{r} \left(1 - \frac{1}{r}\right) \cdot F_3(X) + 1 \cdot \frac{1}{r^2} \cdot F_3(X)$$

$$= \frac{1}{b^2} - \frac{3F_2(X)}{b^2 r} + \frac{3F_2(X)}{br} + \frac{2F_3(X)}{b^2 r^2} - \frac{3F_3(X)}{br^2} + \frac{F_3(X)}{r^2}$$

$$= \frac{1}{b^2} + 3F_2(X) \left(\frac{1}{br} - \frac{1}{b^2 r}\right) + F_3(X) \left(\frac{1}{r^2} - \frac{3}{br^2} + \frac{2}{b^2 r^2}\right)$$

$$= \frac{1}{b^2} + 3F_2(X) \left(\frac{b-1}{b^2 r}\right) + F_3(X) \left(\frac{(b-1)(b-2)}{b^2 r^2}\right) \qquad \square$$

We next show that the all-pairs collision frequency of a collection of independent and identically distributed random variables has a variance that can be expressed in terms of frequency moments. Incidentally, we believe that the following lemma can be simplified and generalized by using results on U-statistics.

**Lemma 3.** *Let $V_1, \ldots, V_n$ be independent random variables such that each $V_i$ has the same distribution as random variable $V$. If*

$$\bar{C} = \frac{2}{n(n-1)} \sum_{i<j} \mathbf{1}\{V_i = V_j\}$$

*then*

$$\mathrm{Var}[\bar{C}] = \frac{2}{n(n-1)} \left(F_2(V) - F_2(V)^2\right) + \frac{4(n-2)}{n(n-1)} \left(F_3(V) - F_2(V)^2\right).$$

*Proof.* Let $C_{ij} = \mathbf{1}\{V_i = V_j\}$. By Bienaymé's identity (Bauer, 2011) we have

$$\mathrm{Var}\left[\sum_{i<j} C_{ij}\right] = \sum_{\substack{i_1 < j_1 \\ i_2 < j_2}} \mathrm{Cov}\left[C_{i_1 j_1}, C_{i_2 j_2}\right].$$

Let $\mathbf{i} = (i_1, i_2)$ and $\mathbf{j} = (j_1, j_2)$. Let $d(\mathbf{i}, \mathbf{j})$ be the number of indices among $i_1, i_2, j_1$ and $j_2$, that are distinct. For example, if $i_1 = i_2 = 1$ and $j_1 = j_2 = 2$ then $d(\mathbf{i}, \mathbf{j}) = 2$. Clearly $d(\mathbf{i}, \mathbf{j}) \le 4$. Also, if $i_1 < j_1$ and $i_2 < j_2$ then $d(\mathbf{i}, \mathbf{j}) \ge 2$. Thus continuing from above we have

$$\sum_{\substack{i_1 < j_1 \\ i_2 < j_2}} \mathrm{Cov}\left[C_{i_1 j_1}, C_{i_2 j_2}\right] = \sum_{\substack{i_1 < j_1 \\ i_2 < j_2 \\ d(\mathbf{i}, \mathbf{j}) = 2}} \mathrm{Cov}[C_{i_1 j_1}, C_{i_2 j_2}] + \sum_{\substack{i_1 < j_1 \\ i_2 < j_2 \\ d(\mathbf{i}, \mathbf{j}) = 3}} \mathrm{Cov}[C_{i_1 j_1}, C_{i_2 j_2}] + \sum_{\substack{i_1 < j_1 \\ i_2 < j_2 \\ d(\mathbf{i}, \mathbf{j}) = 4}} \mathrm{Cov}[C_{i_1 j_1}, C_{i_2 j_2}]$$

We will simplify each term on the right-hand side of the above equation.

Note that $C_{ij}$ is a Bernoulli random variable that is equal to $1$ with probability $\Pr[V_i = V_j]$. Thus by Lemma 1 if $i \ne j$ then $\mathrm{Var}[C_{ij}] = F_2(V) - F_2(V)^2$. If $i_1 < j_1$ and $i_2 < j_2$ and $d(\mathbf{i}, \mathbf{j}) = 2$ then we must have $i_1 = i_2$ and $j_1 = j_2$, which implies

$$\mathrm{Cov}[C_{i_1 j_1}, C_{i_2 j_2}] = \mathrm{Var}[C_{i_1 j_1}] = F_2(V) - F_2(V)^2.$$

Also, if $i_1 < j_1$ and $i_2 < j_2$ and $d(\mathbf{i}, \mathbf{j}) = 3$ then there must exist distinct indices $i, j, k$ such that

$$\begin{aligned}
\mathrm{Cov}[C_{i_1 j_1}, C_{i_2 j_2}] &= \mathrm{E}[C_{i_1 j_1} C_{i_2 j_2}] - \mathrm{E}[C_{i_1 j_1}] \mathrm{E}[C_{i_2 j_2}] \\
&= \mathrm{E}[\mathbf{1}\{V_{i_1} = V_{j_1}\} \mathbf{1}\{V_{i_2} = V_{j_2}\}] - \mathrm{E}[\mathbf{1}\{V_{i_1} = V_{j_1}\}] \mathrm{E}[\mathbf{1}\{V_{i_2} = V_{j_2}\}] \\
&= \mathrm{E}[\mathbf{1}\{V_i = V_j\} \mathbf{1}\{V_j = V_k\}] - \mathrm{E}[\mathbf{1}\{V_{i_1} = V_{j_1}\}] \mathrm{E}[\mathbf{1}\{V_{i_2} = V_{j_2}\}] \\
&= \Pr[V_i = V_j = V_k] - \Pr[V_{i_1} = V_{j_1}] \Pr[V_{i_2} = V_{j_2}] \\
&= F_3(V) - F_2(V)^2. \qquad \qquad \qquad \therefore \text{ Lemma 1}
\end{aligned}$$

Also, if $d(\mathbf{i}, \mathbf{j}) = 4$ then $C_{i_1 j_1}$ is independent of $C_{i_2 j_2}$, and therefore

$$\mathrm{Cov}[C_{i_1 j_1}, C_{i_2 j_2}] = 0.$$

Finally, counting arguments show that

$$|\{(i_1, i_2, j_1, j_2) : i_1 < j_1 \wedge i_2 < j_2 \wedge d(\mathbf{i}, \mathbf{j}) = 2\}| = \frac{n(n-1)}{2}$$

$$|\{(i_1, i_2, j_1, j_2) : i_1 < j_1 \wedge i_2 < j_2 \wedge d(\mathbf{i}, \mathbf{j}) = 3\}| = \frac{n^2(n-1)^2}{4} - \frac{n(n-1)(n-2)(n-3)}{4} - \frac{n(n-1)}{2}$$

$$= n(n-1)(n-2)$$

Putting everything together, we have

$$\begin{aligned} \mathrm{Var}[\bar{C}] &= \mathrm{Var}\left[\frac{2}{n(n-1)} \sum_{i<j} C_{ij}\right] \\ &= \frac{4}{n^2(n-1)^2} \mathrm{Var}\left[\sum_{i<j} C_{ij}\right] \\ &= \frac{4}{n^2(n-1)^2} \left(\frac{n(n-1)}{2}\left(F_2(V) - F_2(V)^2\right) + n(n-1)(n-2)\left(F_3(V) - F_2(V)^2\right)\right) \\ &= \frac{2}{n(n-1)}\left(F_2(V) - F_2(V)^2\right) + \frac{4(n-2)}{n(n-1)}\left(F_3(V) - F_2(V)^2\right). \qquad \square \end{aligned}$$

We next show that the median of a collection of random variables is a good estimate of their common mean, provided that each random variable has small variance. The proof of the following lemma is adapted from Lugosi & Mendelson (2019).

**Lemma 4.** *Let $Z_1, \ldots, Z_n$ be independent random variables such that $\mathrm{E}[Z_i] = \mu$ and $\mathrm{Var}[Z_i] \leq \sigma^2$. If $M$ is the median of $Z_1, \ldots, Z_n$ then*

$$\Pr[|M - \mu| \geq 2\sigma] \leq \exp\left(-\frac{n}{8}\right).$$

*Proof.* Choose any $a > 0$. Let $Y_i = \mathbf{1}\left\{|Z_i - \mu| \geq \sqrt{a}\sigma\right\}$ and $p_i = \mathrm{E}[Y_i]$. We have

$$\begin{aligned} p_i &= \Pr[Y_i = 1] \\ &= \Pr\left[|Z_i - \mu| \geq \sqrt{a}\sigma\right] \\ &\leq \frac{1}{a} \qquad\qquad \because \text{Chebyshev's inequality} \end{aligned}$$

Let $\bar{Y} = \frac{1}{n}\sum_i Y_i$ and $\bar{p} = \frac{1}{n}\sum_i p_i$. Clearly $\mathrm{E}[\bar{Y}] = \bar{p} \leq \frac{1}{a}$. Observe that if $|M - \mu| \geq \sqrt{a}\sigma$ then $\bar{Y} \geq \frac{1}{2}$. Therefore

$$\begin{aligned} \Pr\left[|M - \mu| \geq \sqrt{a}\sigma\right] &\leq \Pr\left[\bar{Y} \geq \frac{1}{2}\right] \\ &\leq \Pr\left[\bar{Y} \geq \bar{p} + \left(\frac{1}{2} - \frac{1}{a}\right)\right] \\ &\leq \exp\left(-2n\left(\frac{1}{2} - \frac{1}{a}\right)^2\right) \qquad \because \text{Hoeffding's inequality} \end{aligned}$$

Setting $a = 4$ proves the lemma. $\qquad \square$

We are ready to prove Theorem 2.

*Proof of Theorem 2.* Let $V$ be defined as in Lemma 2. By the definitions in Mechanism 1 we have for any group $g$

$$\mathrm{E}[\hat{c}_g] = \mathrm{E}\left[\frac{r(b\bar{c}_g - 1)}{b - 1}\right]$$

$$= \left(\frac{br}{b-1}\right)\left(\mathrm{E}\left[\bar{c}_g\right] - \frac{1}{b}\right)$$

$$= \left(\frac{br}{b-1}\right)\left(\frac{2}{m(m-1)}\sum_{\substack{i,j\in I_g \\ i<j}}\mathrm{Pr}\left[v_i = v_j\right] - \frac{1}{b}\right)$$

$$= \left(\frac{br}{b-1}\right)\left(F_2(V) - \frac{1}{b}\right) \qquad\qquad \because \text{Lemma 1}$$

$$= F_2(X) \qquad\qquad \because \text{Lemma 2}$$

and

$$\mathrm{Var}[\hat{c}_g] = \left(\frac{br}{b-1}\right)^2 \mathrm{Var}[\bar{c}_g]$$

$$= \left(\frac{br}{b-1}\right)^2\left(\frac{2}{m(m-1)}\left(F_2(V) - F_2(V)^2\right) + \frac{4(m-2)}{m(m-1)}\left(F_3(V) - F_2(V)^2\right)\right). \quad \because \text{Lemma 3}$$

We can replace $F_2(V)$ and $F_3(V)$ with $F_2(X)$ and $F_3(X)$, respectively, in the above expression using Lemma 2. We have

$$\left(\frac{br}{b-1}\right)^2\left(F_2(V) - F_2(V)^2\right)$$

$$= \left(\frac{br}{b-1}\right)^2\left(\frac{1}{b} + F_2(X)\left(\frac{b-1}{br}\right) - \left(\frac{1}{b} + F_2(X)\left(\frac{b-1}{br}\right)\right)^2\right) \qquad \because \text{Lemma 2}$$

$$= \left(\frac{br}{b-1}\right)^2\left(\frac{1}{b} + F_2(X)\left(\frac{b-1}{br}\right) - \frac{1}{b^2} - \frac{2F_2(X)}{b}\left(\frac{b-1}{br}\right) - F_2(X)^2\left(\frac{b-1}{br}\right)^2\right)$$

$$= \frac{br^2}{(b-1)^2} + F_2(X)\left(\frac{br}{b-1}\right) - \frac{r^2}{(b-1)^2} - 2F_2(X)\left(\frac{r}{b-1}\right) - F_2(X)^2$$

$$= \frac{r^2}{b-1} + F_2(X)\frac{r(b-2)}{b-1} - F_2(X)^2 \tag{8}$$

and

$$\left(\frac{br}{b-1}\right)^2\left(F_3(V) - F_2(V)^2\right)$$

$$= \left(\frac{br}{b-1}\right)^2\left(\frac{1}{b^2} + 3F_2(X)\left(\frac{b-1}{b^2 r}\right) + F_3(X)\left(\frac{(b-1)(b-2)}{b^2 r^2}\right)\right.$$

$$\left. - \left(\frac{1}{b} + F_2(X)\left(\frac{b-1}{br}\right)\right)^2\right) \qquad \because \text{Lemma 2}$$

$$= \left(\frac{br}{b-1}\right)^2\left(\frac{1}{b^2} + 3F_2(X)\left(\frac{b-1}{b^2 r}\right) + F_3(X)\left(\frac{(b-1)(b-2)}{b^2 r^2}\right)\right.$$

$$\left. - \frac{1}{b^2} - \frac{2F_2(X)}{b}\left(\frac{b-1}{br}\right) - F_2(X)^2\left(\frac{b-1}{br}\right)^2\right)$$

$$= 3F_2(X)\left(\frac{r}{b-1}\right) + F_3(X)\left(\frac{b-2}{b-1}\right) - 2F_2(X)\left(\frac{r}{b-1}\right) - F_2(X)^2$$

$$= F_2(X)\left(\frac{r}{b-1}\right) - F_2(X)^2 + F_3(X)\left(\frac{b-2}{b-1}\right) \tag{9}$$

Plugging Eq. (8) and Eq. (9) into the expression for $\mathrm{Var}[\hat{c}_g]$ above we have

$$\mathrm{Var}[\hat{c}_g] = \frac{2}{m(m-1)}\left(\frac{r^2}{b-1} + F_2(X)\frac{r(b-2)}{b-1} - F_2(X)^2\right)$$

$$+ \frac{4(m-2)}{m(m-1)} \left( F_2(X) \left( \frac{r}{b-1} \right) - F_2(X)^2 + F_3(X) \left( \frac{b-2}{b-1} \right) \right)$$
$$= \sigma^2.$$

Thus if $k = 1$ we have

$$\Pr \left[ |\hat{T} - T(X)| \geq \sigma \sqrt{\frac{1}{\delta}} \right] = \Pr \left[ |\hat{c}_1 - \mathrm{E}[\hat{c}_1]| \geq \sigma \sqrt{\frac{1}{\delta}} \right] \leq \delta$$

by Chebyshev's inequality, and if $k = 8 \log \frac{1}{\delta}$ we have

$$\Pr \left[ |\hat{T} - T(X)| \geq 2\sigma \right] = \Pr \left[ |\hat{m} - F_2(X)| \geq 2\sigma \right] \leq \delta$$

by Lemma 4. $\qquad \square$

## D  PROOF OF THEOREM 3

We make use of the lower bound for local differential privacy introduced by Duchi et al. (2016) which is relying on a privatized version of Le Cam's two point method. Accordingly, we construct a pair of problem instances $\mathbf{p}$ and $\mathbf{p}'$ for which $d_C(\mathbf{p}_0, \mathbf{p}_1) = |C(\mathbf{p}) - C(\mathbf{p}')| \geq \Omega(\tau)$ and at the same time $d_K L(\mathbf{p}_0, \mathbf{p}_1) \in \Theta(\tau^2)$ which can be plugged into the privatized lower and results in the optimality of Mechanism 1 in terms of $\alpha$ and $\epsilon$ in minimax sense. For doing so, let us define

$$\mathbf{p}_0 = \left( \frac{1}{2(K-1)}, \dots, \frac{1}{2(K-1)}, \frac{1}{2} \right) \quad \text{and} \quad \mathbf{p}_1 = \left( \frac{1-\tau}{2(K-1)}, \dots, \frac{1-\tau}{2(K-1)}, \frac{1+\tau}{2} \right) \tag{10}$$

The KL divergence between $\mathbf{p}_0$ and $\mathbf{p}_1$ is

$$d_{\mathrm{KL}}(\mathbf{p}_0, \mathbf{p}_1) = \frac{1}{2} \log \frac{1}{1 - \tau^2} = \Theta(\tau^2)$$

and the absolute difference between their collision probability is

$$d_C(\mathbf{p}_0, \mathbf{p}_1) = |C(\mathbf{p}_0) - C(\mathbf{p}_1)| = \frac{\tau}{2} \left( 1 + \left( \frac{\tau}{2} - \frac{1}{2(K-1)} \right) \right) \geq \tau/2$$

The lower bound of Duchi et al. (2016) readily implies the following Corollary.

**Corollary 1.** *Suppose that $\theta$ is an estimator of $C(\mathbf{p})$ which gets $n$ observations from an $\alpha$-locally differential private channel $Q$ with $\alpha \in [0, 23/35]$, i.e. channel $Q$ is a conditional probability distribution which maps each observation $x_i$ to a probability distribution on some finite discrete domain $\mathcal{Z}$. We will denote the privatized data by $Z_i \sim Q(.|x_i)$. Then for any pair of distributions $\mathbf{p}_0$ and $\mathbf{p}_1$ such that $d_C(\mathbf{p}_0, \mathbf{p}_1) \geq \tau/2$, then it holds*

$$\inf_Q \inf_\theta \sup_{\mathbf{p}} \mathbb{E}_{Q,\mathbf{p}} [d_C(\mathbf{p}, \theta(Z_1, \dots, Z_n))] \geq \frac{\tau}{4} \left( 1 - \sqrt{2\alpha^2 n d_{KL}(\mathbf{p}_0, \mathbf{p}_1)} \right)$$

Corollary 1 applied to the the pair of distribution defined (10) with $\tau = 1/(\alpha\sqrt{n})$ implies that Mechanism 1 is minimax optimal in terms of $\epsilon$ and $\alpha$ by achieving a sample complexity that is $O(1/(\alpha\sqrt{n}))$. Note that we do not focus on communication complexity here, so the domain $\mathcal{Z}$ can be arbitrary large but finite.

## E  PROOF OF THEOREM 4

We have

$$\sum_v \hat{n}_v^2 = \sum_v \left( \sum_i \mathbf{1}\{v_i = v\} \right)^2$$
$$= \sum_v \sum_{i,j} \mathbf{1}\{v_i = v\} \mathbf{1}\{v_j = v\}$$

$$= \sum_v \sum_{i,j} \mathbf{1}\left\{v_i = v_j = v\right\}$$

$$= \sum_{i,j} \mathbf{1}\left\{v_i = v_j\right\}$$

$$= m + 2\sum_{i<j} \mathbf{1}\left\{v_i = v_j\right\}$$

and rearranging proves the theorem.

## F    PROOF OF THEOREM 5

First let us define

$$U_m = \frac{2}{m(m-1)} \sum_{i=1}^{m} \sum_{j=1}^{i-1} \mathbf{1}\left\{X_i = X_j\right\} \tag{11}$$

based on $\{X_1, \ldots, X_m\}$. The sequence $U_1, U_2, \ldots$ are dependent sequences, since each of them depends on all previous observations, thus we shall apply a decoupling technique to obtain a martingale sequence which we can use in a sequential test. Based on $U_m$, let us define

$$\bar{U}_m := \sum_{i=1}^{m} \sum_{j=1}^{i-1} g_{\mathbf{p}}(X_i, X_j)$$

with

$$g_{\mathbf{p}}(X_i, X_j) = \mathbf{1}\left\{X_i = X_j\right\} - \Pr\left(X_i = X_j | X_i\right) - \Pr\left(X_i = X_j | X_j\right) + C(\mathbf{p}) \ .$$

This decoupling technique is motivated by Theorem 8.1.1 of Tsybakov (2008) since the kernel function $g$ has became centered and degenerated, i.e. $\mathrm{E}\left[g_{\mathbf{p}}(X_i, X_j)|X_j\right] = \mathrm{E}\left[g_{\mathbf{p}}(X_i, X_j)|X_i\right] = 0$ which implies that $\bar{U}_n$ is a zero-mean martingale with $n \geq 2$ as follows.

**Lemma 5.** $\bar{U}_2, \bar{U}_3, \ldots$ *is a discrete-time martingale the filtration of which is defined* $\mathcal{F}_t = \{X_1, \ldots, X_m\}$ *and for all* $m$,

$$\mathrm{E}\left[Y_m(\mathbf{p})|\mathcal{F}_{m-1}\right] = 0$$

*where* $Y_m(\mathbf{p}) = \sum_{i=1}^{m-1} g_{\mathbf{p}}(X_m, X_i)$ *if* $m \geq 2$ *and* $Y_1 = 0$.

*Proof.* This decoupling is motivated by Theorem 8.1.1 of de la Peña & Giné (1999). First note that $\mathrm{E}\left[g_{\mathbf{p}}(X_i, X_j)|X_j\right] = \mathrm{E}\left[g_{\mathbf{p}}(X_i, X_j)|X_i\right] = 0$ by construction. This implies that

$$\mathrm{E}\left[Y_m(\mathbf{p})|\mathcal{F}_{m-1}\right] = \sum_{i=1}^{m-1} \mathrm{E}\left[g_{\mathbf{p}}(X_m, X_i)|\mathcal{F}_{m-1}\right] = \sum_{i=1}^{m-1} \mathrm{E}\left[g_{\mathbf{p}}(X_m, X_i)|X_i\right] = 0$$

for all $m \geq 2$. Since $\bar{U}_m = \sum_{i=1}^{m} Y_i(\mathbf{p})$, it holds that

$$\mathrm{E}\left[\bar{U}_m|\mathcal{F}_{m-1}\right] = \bar{U}_{m-1} + \mathrm{E}\left[Y_m(\mathbf{p})|\mathcal{F}_{m-1}\right] = \bar{U}_{m-1} \ .$$

Finally, it is straightforward that $\mathrm{E}\left[|Y_m(\mathbf{p})|\right] < \infty$ which implies that $\bar{U}_2, \bar{U}_3, \ldots$ is a discrete-time martingale by definition. □

The empirical sequence is $\bar{u}_m = \sum_{i=1}^{m} y_m(\mathbf{p})$ with

$$y_j(\mathbf{p}) = \sum_{i=1}^{m-1} \mathbf{1}\left\{x_i = x_j\right\} - \sum_{i=1}^{m-1} p_{x_i} - (m-1)p_{x_j} + (m-1)C(\mathbf{p})$$

which is a realization of a martingale with bounded difference such that $|\bar{U}_k - \bar{U}_{k-1}| = |Y_k| \leq 4m$ and $y_1(\mathbf{p}) = 0$. However we cannot compute the empirical sequence, since the parameters of distribution are not known. As a remedy, we further decompose $\bar{U}_n$ as the sum of two sequences based on the observation that

$$\mathbb{E}\left[p_{X_i}\right] = \sum_i p_{x_i}^2 = C(\mathbf{p})$$

which implies that $\sum_{i=1}^{m}(p_{X_i} - C(\mathbf{p}))$ which is again a zero-mean martingale sequence with the same filtration $\mathcal{F}_m$ such that the difference $|p_{X_i} - C(X)| < 1$ for all $i$. This motivates the following decomposition of $\bar{U}_n$ as

$$Y_j(\mathbf{p}) = \underbrace{\sum_{i=1}^{j-1} \mathbf{1}\{X_i = X_j\} - 2(j-1)C(\mathbf{p})}_{T_j(\mathbf{p})} + \underbrace{2(j-1)C(\mathbf{p}) - \sum_{i=1}^{j-1} p_{X_i} - (j-1)p_{X_j}}_{E_j(\mathbf{p})}$$

Note that $T_m(\mathbf{p})$ can be computed, and it is a zero-mean martingale sequence up to an error term $E_n(\mathbf{p})$ which we cannot be computed, since the parameters of the underlying distribution $\mathbf{p}$ is not available to the tester. Also note that $T_m(\mathbf{p})$ is a centralized version of $\bar{U}_m$ defined in (11). More detailed, we have that

$$\frac{2}{m(m-1)} \sum_{i=1}^{m} T_m(\mathbf{p}) = U_m - C(\mathbf{p})$$

which means that Algorithm 2 uses the sequence of $U_1, \ldots, U_m$ as test statistic which was our point of departure. Now we will focus on $E_m(\mathbf{p})$ and how it can be upper bounded.

Further note that $E_n(\mathbf{p})$ can be again decomposed into sequence of sums of zero mean-mean terms which we can upper bound with high probability. Due to the construction, it holds that

$$\sum_{i=1}^{m} Y_i(\mathbf{p}) = \sum_{i=1}^{m} T_i(\mathbf{p}) + \sum_{i=1}^{m} E_i(\mathbf{p})$$

We can apply the time uniform confidence interval of Howard et al. (2021) to the lhs which implies that it holds that

$$\Pr\left[\forall m \in \mathbb{N} : \left|\frac{2}{m(m-1)} \sum_{i-1}^{m} Y_i(\mathbf{p})\right| \geq \phi(i, \delta)\right] \leq \delta \ . \tag{12}$$

if the data is generated from $\mathbf{p}$ where

$$\phi(i, \delta) = 1.7\sqrt{\frac{\log\log i + 0.72\log(10.4/\delta)}{i}} \ .$$

Note that the confidence interval of Howard et al. (2021) applies to the sum of discrete time martingales where each term is sub-Gaussian. This also applies to $Y_i(\mathbf{p}$, since it is a bounded random variable.

Next we upper bound $\sum_i E_i(\mathbf{p})$. For doing so, we decompose each term as

$$E_i(\mathbf{p}) = \sum_{j=1}^{i-1} (F_2(\mathbf{p}) - p_{X_j}) + (i-1)(F_2(\mathbf{p}) - p_{X_i})$$

which implies

$$\begin{aligned}
\sum_{i-1}^{m} E_i(\mathbf{p}) &= \sum_{i=1}^{m} \sum_{j=1}^{i-1} (F_2(\mathbf{p}) - p_{X_j}) + \sum_{i=1}^{m} (i-1)(F_2(\mathbf{p}) - p_{X_i}) \\
&= \sum_{i=1}^{m-1} (m-i)(F_2(\mathbf{p}) - p_{X_i}) + \sum_{i=1}^{m} (i-1)(F_2(\mathbf{p}) - p_{X_i}) \\
&= m \sum_{i=1}^{m} (F_2(\mathbf{p}) - p_{X_i})
\end{aligned}$$

Apply the time uniform confidence interval of Howard et al. (2021) to $E_i(\mathbf{p})$, we have that

$$\Pr\left[\forall m \in \mathbb{N} : \left|\frac{2}{m(m-1)} \sum_{i-1}^{m} E_i(\mathbf{p})\right| \geq \phi(i, \delta)\right] \leq \delta \ . \tag{13}$$

Due to union bound, we can upper bound the difference of $T_i(\mathbf{p})$ and $Y_i(\mathbf{p})$ using (13) and (12) as

$$\frac{2}{m(m-1)}\left|\sum_{i=1}^{m}Y_i(\mathbf{p}) - \sum_{i=1}^{m}T_i(\mathbf{p})\right| \leq 2\phi(i,\delta/2)$$

with probability at least $1-\delta$ for all $m$ even if $m$ is a random variable that depends on $X_1,\ldots,X_m$. This implies that if the observations is generated from a distribution with parameters $\mathbf{p}$, then $\frac{2}{m(m-1)}T_i(\mathbf{p})$ stays close to zero, including all distribution $\mathbf{p}_0$ such that $C(\mathbf{p}_0) = c_0$. This implies the correctness of Algorithm 2.

Finally note that

$$\left|\frac{2}{m(m-1)}\sum_{i=1}^{m}Y_m(\mathbf{p}) - \frac{2}{m(m-1)}\sum_{i=1}^{m}Y_i(\mathbf{p}_0)\right| = |C(\mathbf{p}) - \underbrace{C(\mathbf{p}_0)}_{=c_0}|$$

for any $\mathbf{p}_0$ such that $C(\mathbf{p}_0) = c_0$ which implies the sample complexity bound. This concludes the proof.

## G   PROOF OF THEOREM 6

Before we proof the lower bound, we need to get a better understating of the relation of the total variation distance and $d_C(\mathbf{p},\mathbf{p}') = |C(\mathbf{p}) - C(\mathbf{p}')|$

### G.1   TOTAL VARIATION DISTANCE

The *total variation distance* between random variables $X$ and $Y$ is defined

$$|X - Y| = \frac{1}{2}\sum_z |\Pr[X = z] - \Pr[Y = z]|$$

where the sum is over the union of the supports of $X$ and $Y$.

**Theorem 7.** *For any $X$ and $Y$*

$$|C(X) - C(Y)| \leq 6\,|X - Y|.$$

*Proof.* Let $z_1, z_2, \ldots$ be an enumeration of the union of the supports of $X$ and $Y$. Let $p_i = \Pr[X = z_i]$ and $q_i = \Pr[Y = z_i]$.

Assume without loss of generality $C(X) \leq C(Y)$. It suffices to prove $C(Y) \leq C(X) + 6|X - Y|$. Let $\delta_i = p_i - q_i$. We have

$$\begin{aligned}
C(X) &= \sum_i p_i^2 \\
&= \sum_i (q_i + \delta_i)^2 \\
&= \sum_i q_i^2 + 2\sum_i q_i\delta_i + \sum_i \delta_i^2 \\
&\leq \sum_i q_i^2 + 2\sum_i q_i|\delta_i| + \sum_i \delta_i^2 \\
&\leq \sum_i q_i^2 + 2\sum_i |\delta_i| + \sum_i \delta_i^2 \\
&\leq \sum_i q_i^2 + 2\sum_i |\delta_i| + \sum_i |\delta_i| \\
&= C(Y) + 6|X - Y|
\end{aligned}$$

and rearranging completes the proof. □

## G.2 LOWER BOUND

Based on of Lemma A.1 due to Oufkir et al. (2021), one can lower bound the stopping time of any sequential testing algorithm in expectation. Note that this lower bound readily applies to our setup and implies a lower bound for the expected sample complexity which is

$$\frac{\log 1/3\delta}{d_{\text{KL}}(\mathbf{p}, \mathbf{p}')} \tag{14}$$

where

$$d_C(\mathbf{p}, \mathbf{p}') = |C(\mathbf{p}) - C(\mathbf{p}')| = \epsilon$$

In addition to this, the following Lemma lower bounds the sensitivity of KL divergence in terms of collision probability.

**Lemma 6.** *For any random variables $X$ and $X'$ with parameters $\mathbf{p}$ and $\mathbf{p}$, it holds*

$$d_C(\mathbf{p}, \mathbf{p}')^2 \leq 18d_{KL}(\mathbf{p}, \mathbf{p}')$$

*Proof.* Pinsker's inequality and Theorem 7 implies this result. □

Lemma 6 applied to (14) implies that Theorem 5 achieves optimal sample complexity, since for any distribution for which

$$C(\mathbf{p}_0) = c_0$$

and

$$d_C(\mathbf{p}, \mathbf{p}') = \epsilon$$

the expected sample complexity of any tester is lower bounded by

$$\frac{\log 1/3\delta}{\epsilon^2}$$

This concludes the proof.

## H BATCH TESTERS USED IN THE EXPERIMENTS

In the experimental study, we used two batch testers as baseline. Each of these testers are based on learning algorithm which means that using a learning algorithm, the collision probability is estimated with an additive error $\epsilon/2$ and then one can decide whether the true collision probability is close to $c_0$ or not. This approach is caller *testing-by-learning*. In this section, we present exact sample complexity bound for these batch testers and in addition to this, we show that these approaches are optimal in minimax sense for testing collision probability for discrete distributions. In this section we present the following results:

- We start by presenting a minmax lower bound for the batch testing problem which is $\Omega(\epsilon^{-2})$. In addition, we also show that the same lower bound applies to learning.

- In Subsection H.2, we consider two estimators, i.e. plug-in and U-statistic, and we compute their sample complexity upper bound that are of order $\epsilon^{-2}$ and they differ only in constant. These are presented in Theorem 8 and 9, respectively.

- In Subsection H.3, we present the testing-by-learning approach and discuss that the plug-in estimator is minmax optimal on a wide range of parameters.

## H.1 LOWER BOUND FOR ESTIMATION AND TESTING

To construct lower bound for estimation and testing we consider the pair of distributions defined in (10) with $\tau = \epsilon$. In this case, we obtain two distributions such that $d_{\text{KL}}(\mathbf{p}_0, \mathbf{p}_1) = \Theta(\epsilon^2)$ and $d_C(\mathbf{p}_0, \mathbf{p}_1) \geq \epsilon/2$. Then estimator lower bound can be obtained based on LeCam's theorem (See Appendix I.1) which is $\Theta(1/\epsilon^2)$ as follows.

**Corollary 2.** *For any estimator $\hat{\theta}_n$ for Collision probability $F_2(\mathbf{p})$ based on $n \in o(1/\epsilon^2)$, there exist a discrete distribution $\mathbf{p}$ for which*

$$\mathbb{E}_P\left[\left|\hat{\theta}_n(\mathcal{D}_n) - F_2(\mathbf{p})\right|\right] \geq C \cdot \epsilon$$

*where $C > 0$ does not depend on the distribution $\mathbf{p}$.*

One can show a similar lower bound for testing using Neyman-Pearson lemma. We refer the reader to Section 3.1 of Canonne (2022b) for more detail. We recall this result here with $d_C$.

**Corollary 3.** *Let $f$ an $(\epsilon, \delta)$-tester with sample complexity $n$. Then for any pair of distributions $\mathbf{p}_0$ and $\mathbf{p}_1$ such that $d_C(\mathbf{p}_0, \mathbf{p}_1) = \epsilon$, it holds that*

$$1 - 2\delta \leq d_{TV}(\mathbf{p}_0^{\otimes n}, \mathbf{p}_1^{\otimes n})$$

*where $p^{\otimes n}$ is the $n$ times product distribution from $\mathbf{p}$.*

Using Pinsker's inequality it results in that

$$d_{\text{TV}}(\mathbf{p}_0^{\otimes n}, \mathbf{p}_1^{\otimes n})^2 \leq \frac{1}{2}d_{\text{KL}}(\mathbf{p}_0^{\otimes n}, \mathbf{p}_1^{\otimes n}) = \frac{n}{2}d_{\text{KL}}(\mathbf{p}_0, \mathbf{p}_1) \ .$$

Accordingly, since we already constructed a pair of distributions for which $d_2(\mathbf{p}_0, \mathbf{p}_1) = \epsilon$ and $d_{\text{KL}}(\mathbf{p}_0, \mathbf{p}_1) = \Omega(\epsilon^2)$, the sample compelxity lower bound for testing is also $\Omega(1/\epsilon^2)$.

### H.2 Plug-in estimator versus U-statistic estimator

The first estimator is the plug-in estimator which estimates the distribution $\mathbf{p}$ by the normalized empirical frequencies $\widehat{\mathbf{p}} := \widehat{\mathbf{p}}(\mathcal{D}_m)$ and then the estimator is computed as

$$C(\widehat{\mathbf{p}}) = F_2(\widehat{\mathbf{p}}) = \sum_{i=1}^{K} \widehat{p}_i^2$$

In this section, we will other frequency moments of discrete distributions, therefore we will use $F_k(\mathbf{p})$ as the frequency moment of order $k$, which is the collision probability with $k = 2$.

The plug-in estimator is well-understood in the general case via lower and upper bound that are presented in Acharya et al. (2014). Here we recall an additive error bound under Poissonization which assumes that the sample size is chosen as $M \sim \text{Poi}(m)$ and the data is then $\mathcal{D}_M$.

**Theorem 8.** *If*

$$m \geq \max\left\{\frac{1600F_{3/2}(\mathbf{p})^2}{\epsilon^2}, \frac{8}{\epsilon^2}\log\frac{2}{\delta}\right\} = \frac{8}{\epsilon^2} \cdot \max\left\{200 \cdot F_{3/2}(\mathbf{p})^2, \log\frac{2}{\delta}\right\} \ .$$

*then*

$$\mathbb{P}\left(|F_2(\widehat{\mathbf{p}}(\mathcal{D}_M)) - T_2(X)| \geq \epsilon\right) \leq \delta$$

*where the dataset $\mathcal{D}_M$ is sampled with sample size $M \sim \text{Poi}(m)$.*

*Proof.* Based on Theorem 9 of Acharya et al. (2014), the bias of the estimator with Poissonization is

$$|\mathbb{E}\left[F_2(\widehat{\mathbf{p}}(\mathcal{D}_M))\right] - T_2(X)| \leq \frac{8}{m} + \frac{10}{\sqrt{m}}F_{3/2}(\mathbf{p})$$

and its variance is

$$\mathbb{V}\left[F_2(\widehat{\mathbf{p}}(\mathcal{D}_M))\right] \leq \frac{64}{m^3} + \frac{4 \cdot 17}{\sqrt{m}}F_{7/2}(\mathbf{p}) \ .$$

Thus

$$\mathbb{P}\left(|F_2(\widehat{\mathbf{p}}(\mathcal{D}_M)) - T_2(X)| \geq \epsilon\right) = \mathbb{P}\left(|F_2(\widehat{\mathbf{p}}(\mathcal{D}_M)) - \mathbb{E}\left[F_2(\widehat{\mathbf{p}}(\mathcal{D}_M))\right] + \mathbb{E}\left[F_2(\widehat{\mathbf{p}}(\mathcal{D}_M))\right] - T_2(X)| \geq \epsilon\right)$$
$$\leq \mathbb{P}\left(|F_2(\widehat{\mathbf{p}}(\mathcal{D}_M)) - \mathbb{E}\left[F_2(\widehat{\mathbf{p}}(\mathcal{D}_M))\right]| \geq \epsilon - |\mathbb{E}\left[F_2(\widehat{\mathbf{p}}(\mathcal{D}_M))\right] - T_2(X)|\right)$$

where we applied the triangle inequality. Therefore if $m$ is big enough, then it holds that

$$\frac{8}{m} + \frac{10}{\sqrt{m}}F_{3/2}(\mathbf{p}) \leq \frac{\epsilon}{2} \tag{15}$$

and also holds

$$\mathbb{P}\left(|F_2(\widehat{\mathbf{p}}(\mathcal{D}_M)) - \mathbb{E}\left[F_2(\widehat{\mathbf{p}}(\mathcal{D}_M))\right]| \geq \epsilon/2\right) \leq \delta \tag{16}$$

thus the statement in the theorem holds. What remains is to compute a lower bound for $m$. (15) holds if

$$m \geq \max\left\{\frac{32}{\epsilon}, \frac{1600 F_{3/2}(\mathbf{p})^2}{\epsilon^2}\right\} .$$

Based on Bernstein's inequality, see Theorem 11 in Appendix I, (16) hold if

$$m \geq \max\left\{\frac{8\log\frac{2}{\delta}}{\epsilon^2}, \frac{4736 \cdot \sqrt[4]{\log\frac{2}{\delta}}}{\sqrt{\epsilon}}, \frac{6528\sqrt[3]{F_{7/2}(\mathbf{p})\log\frac{2}{\delta}}}{\epsilon^{4/3}}\right\}$$

which concludes the proof. To simplify the last terms, alternatively we can apply Hoeffding's inequality which yields that (16) holds whenever

$$m \geq \frac{8}{\epsilon^2}\log\frac{2}{\delta} .$$

Finally note that $32/\epsilon \leq 8/\epsilon^2\log 2/\delta$ for any $\epsilon, \delta \in (0, 1]$ which concludes the proof. $\qquad\square$

**Theorem 9.** *If*

$$m \geq \max\left\{\frac{32(F_3(X) - F_2(X)^2)}{\epsilon^2}\ln\frac{4}{\delta}, \frac{128 + 1/6}{\epsilon}\ln\frac{4}{\delta}\right\}$$

*then*

$$\mathbb{P}\left(|F_2(X) - U(\mathcal{D}_m)| \geq \epsilon\right) \leq \delta$$

*Proof.* Based on Theorem 9 of Acharya et al. (2014), the bias of the estimator with Poissonization is

$$|\mathbb{E}\left[F_2(\widehat{\mathbf{p}}(\mathcal{D}_M))\right] - T_2(X)| \leq \frac{8}{m} + \frac{10}{\sqrt{m}}F_{3/2}(\mathbf{p})$$

and its variance is

$$\mathbb{V}\left[F_2(\widehat{\mathbf{p}}(\mathcal{D}_M))\right] \leq \frac{64}{m^3} + \frac{4 \cdot 17}{\sqrt{m}}F_{7/2}(\mathbf{p}) .$$

Thus

$$\mathbb{P}\left(|F_2(\widehat{\mathbf{p}}(\mathcal{D}_M)) - T_2(X)| \geq \epsilon\right) = \mathbb{P}\left(|F_2(\widehat{\mathbf{p}}(\mathcal{D}_M)) - \mathbb{E}\left[F_2(\widehat{\mathbf{p}}(\mathcal{D}_M))\right] + \mathbb{E}\left[F_2(\widehat{\mathbf{p}}(\mathcal{D}_M))\right] - T_2(X)| \geq \epsilon\right)$$
$$\leq \mathbb{P}\left(|F_2(\widehat{\mathbf{p}}(\mathcal{D}_M)) - \mathbb{E}\left[F_2(\widehat{\mathbf{p}}(\mathcal{D}_M))\right]| \geq \epsilon - |\mathbb{E}\left[F_2(\widehat{\mathbf{p}}(\mathcal{D}_M))\right] - T_2(X)|\right)$$

where we applied the triangle inequality. Therefore if $m$ is big enough, then it holds that

$$\frac{8}{m} + \frac{10}{\sqrt{m}}F_{3/2}(\mathbf{p}) \leq \frac{\epsilon}{2} \tag{17}$$

and also holds

$$\mathbb{P}\left(|F_2(\widehat{\mathbf{p}}(\mathcal{D}_M)) - \mathbb{E}\left[F_2(\widehat{\mathbf{p}}(\mathcal{D}_M))\right]| \geq \epsilon/2\right) \leq \delta \tag{18}$$

thus the statement in the theorem holds. What remains is to compute a lower bound for $m$. (17) holds if

$$m \geq \max\left\{\frac{32}{\epsilon}, \frac{1600 F_{3/2}(\mathbf{p})^2}{\epsilon^2}\right\} .$$

Based on Bernstein's inequality, see Theorem 11 in Appendix I, (18) hold if

$$m \geq \max\left\{\frac{8\log\frac{2}{\delta}}{\epsilon^2}, \frac{4736 \cdot \sqrt[4]{\log\frac{2}{\delta}}}{\sqrt{\epsilon}}, \frac{6528\sqrt[3]{F_{7/2}(\mathbf{p})\log\frac{2}{\delta}}}{\epsilon^{4/3}}\right\}$$

which concludes the proof. To simplify the last terms, alternatively we can apply Hoeffding's inequality which yields that (18) holds whenever

$$m \geq \frac{8}{\epsilon^2}\log\frac{2}{\delta} .$$

Finally note that $32/\epsilon \leq 8/\epsilon^2\log 2/\delta$ for any $\epsilon, \delta \in (0, 1]$ which concludes the proof. $\qquad\square$

Note that as soon as $(F_3(X) - F_2(X)^2)/5 \leq \epsilon$, the second term of the sample complexity of Theorem 9 becomes dominant, and thus the sample complexity in these parameter regime is $O(\ln(1/\delta)/\epsilon)$. In addition to this, it is easy to see that the first tern of the sample complexity is zero when $X$ is distributed uniformly.

### H.3 TESTING BY LEARNING

Testing by learning consists of estimating the parameter itself with a small additive error which allows us to distinguish between null $H_0$ and alternative hypothesis $H_1$. This approach had been found to be optimal in several testing problem Busa-Fekete et al. (2021), as it is also optimal in this case based on the lower bound presented in the previous section. We considered several estimators for Collision entropy which can be used in a batch testing setup by setting the sample size so as the additive error of the estimator is smaller than $\epsilon/2$. In this way, we can distinguish between $H_0$ and $H_1$ as expected. The confidence interval of each estimator does depend on some frequency moment of the underlying distribution which can be upper worst case upper bounded. For example, the plug-n estimator sample complexity $m$ is $1600/\epsilon^2$ if $e^{-199} \leq \delta$.

## I TECHNICAL TOOLS

### I.1 LECAM'S LOWER BOUND

Let $\hat{\theta}_n = \hat{\theta}(x_1, \dots, x_n)$ such that $\hat{\theta}_n : (\Sigma^d)^n \mapsto \mathbb{R}$ be an estimator using $n$ samples.

**Theorem 10.** *[Le Cam's theorem] Let $\mathcal{P}$ be a set of distributions. Then, for any pair of distributions $P_0, P_1 \in \mathcal{P}$, we have*

$$\inf_{\hat{\theta}} \max_{P \in \mathcal{P}} \mathbb{E}_P \left[ d(\hat{\theta}_n(P), \theta(P)) \right] \geq \frac{d(\theta(P_0), \theta(P_1))}{8} e^{-n d_{KL}(P_0, P_1)},$$

*where $\theta(P)$ is a parameter taking values in a metric space with metric $d$, and $\hat{\theta}_n$ is the estimator of $\theta$ based on $n$ samples.*

### I.2 BERSNTEIN'S BOUND

The following form of Bernstein's bound can be derived from Theorem 1.4 of Dubhashi & Panconesi (2009).

**Theorem 11.** *(Bernstein's bound) Let $X_1, \dots, X_n$ be i.i.d. random variables, and $\forall i \in [n], |X_i - \mathbb{E}[X_i]| \leq b$ and $\mathbb{E}[X_i] = \mu$. Let $\sigma^2 = \mathbb{V}[X_i]$. Then with probability at least $1 - \delta$ it holds that*

$$\left| \frac{1}{n} \sum_{i=1}^n X_i - \mu \right| \leq \sqrt{\frac{4\sigma^2 \ln \frac{2}{\delta}}{n}} + \frac{4b \ln \frac{2}{\delta}}{3n} \ .$$

## J COMPARISON TO DASKALAKIS & KAWASE (2017)

Daskalakis & Kawase (2017) described a sequential testing algorithm that can be adapted to collision probability testing. However, their approach has two major disadvantages relative to Algorithm 2, both of which lead to much higher sample complexities in practice. First, their approach is based on a simple "doubling trick": They repeatedly invoke a non-sequential testing algorithm on subsequences of samples with successively smaller values of the error tolerance $\epsilon$, and stop when the testing algorithm rejects. This is a wasteful use of samples compared to our approach, as stopping cannot occur within a subsequence, and everything learned from previous subsequences is discarded. Second, applying their approach to collision probability testing requires partitioning the $n$ samples into $\frac{n}{2}$ disjoint pairs, so that the observed collisions are independent of each other. By contrast, our approach uses observed collisions among all $\binom{n}{2}$ pairs of samples to estimate collision probability, which significantly complicates the theoretical analysis, but leads to better empirical performance.

We ran experiments comparing the two algorithms on the power law distribution ($p_i \propto 1/i$) and exponential distribution ($p_i \propto \exp(-i)$), with the results depicted in Figure 4. We found that as each

tester's null hypothesis approaches the true collision probability, the empirical sample complexity of Daskalakis & Kawase (2017)'s algorithm became much larger than the empirical sample complexity of Algorithm 2.

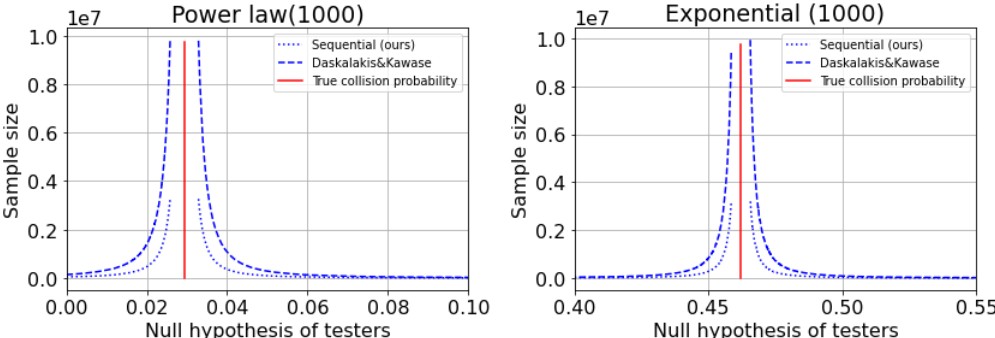

Figure 4: Sample complexity of our sequential tester (Algorithm 2) compared to the sample complexity of Daskalakis & Kawase (2017)'s sequential tester adapted for collision probability testing.

## K  FAILURE OF REDUCTION TO DISTRIBUTION ESTIMATION

In Section 6 we showed empirically that our method for privately estimating the collision probability of a distribution outperforms the indirect method of privately estimating the distribution itself and then computing the collision probability of the estimated distribution. In this section we prove a theoretical separation between these two methods, by showing that the reduction does not work in general. Specifically, we show that even if the private distribution estimation algorithm has the optimal sample complexity, using it as a subroutine for collision probability estimation may require a number of samples that depends on the support size of the distribution. By contrast, the sample complexity of our method is independent of support size (see Theorem 2).

Let $[k] = \{1, \ldots, k\}$ be the sample space. Let $\Delta_k$ be the set of all distributions on $[k]$. Let $A : [k]^n \to \Delta_k$ denote an algorithm that inputs $n$ samples, one per user, and outputs an estimated distribution. An $(\alpha, \beta)$-local differentially private algorithm $A^*$ is $(\alpha, \beta)$-*minimax optimal* if

$$A^* = \arg\min_A \max_{\mathbf{p} \in \Delta^k} \mathrm{E}_{(x_1, \ldots, x_n) \sim \mathbf{p}^n}[\|A(x_1, \ldots, x_n) - \mathbf{p}\|_1]$$

where the minimization is over all $(\alpha, \beta)$-local differentially private algorithms. Recall from Section 3 that $C(\mathbf{p}) = \sum_{i=1}^k p_i^2$ denotes the collision probability of a distribution $\mathbf{p} \in \Delta_k$.

**Theorem 12.** *There exists an algorithm $A^* : [k]^n \to \Delta_k$ that is $(\alpha, 0)$-minimax optimal for all $\alpha \geq \log k$ and a distribution $\mathbf{p} \in \Delta^k$ such that if each $x_i \in [k]$ is drawn independently from $\mathbf{p}$ then*

$$\mathrm{E}\left[|C(A(x_1, \ldots, x_n)) - C(\mathbf{p})|\right] \geq \Omega\left(\min\left\{1, \frac{k}{\sqrt{n}}\right\}\right).$$

*Proof.* It is known that $k$-ary randomized response is $(\alpha, 0)$-minimax optimal for all $\alpha \geq \log k$ (Acharya et al., 2019b). In this algorithm, each user $i$ reports their true value $x_i$ with probability $\frac{e^\varepsilon}{e^\varepsilon + k - 1}$, and otherwise reports another value chosen uniformly at random. The estimated distribution $\tilde{\mathbf{p}}$ is defined by $\tilde{p}_i = b(\hat{p}_i - a)$, where $\hat{\mathbf{p}}$ is the empirical distribution and $a = \frac{k-1}{e^\varepsilon + k - 1}$ and $b = \frac{e^\varepsilon + k - 1}{e^\varepsilon - k + 1}$ serve to debias the noise introduced by the randomized response. Now suppose the true distribution $\mathbf{p}$ is concentrated on a single element. It follows straightforwardly from some algebra that for large $n$ we have $\mathrm{E}[|C(\tilde{\mathbf{p}}) - C(\mathbf{p})|] \geq \frac{b}{\sqrt{n}}$. Noting that $b = \Omega(k)$ when $\alpha = \log k$ completes the proof. □

By contrast, our method for estimating collision probability is $(\alpha, \beta)$-differentially private, and if $\alpha = \log k$ it achieves $O\left(\sqrt{\frac{\log \frac{1}{\beta}}{n}}\right)$ error (see Theorem 2).

