# OpenReview forum: "Near-optimal algorithms for private estimation and sequential testing of collision probability"
_ICLR.cc/2024/Conference — Submitted to ICLR 2024_

### Official Review · Reviewer_cQaE · 2023-10-14

**Soundness:** 2 fair
**Presentation:** 2 fair
**Contribution:** 2 fair
**Rating:** 3
**Confidence:** 4

**Summary:**

This paper studies the problem of estimating, given samples from a discrete distributions, the collision probability of the distribution to within an *additive* (constant) factor. The collision probability is the probability that a pair of samples from the distribution collides (equals the same value), and is essentially equivalent to the well-studied second moment of the distribution. For a distribution with support size
$k$ and probabilities $p_1, \ldots, p_k$, the collision probability is $\sum_{i=1}^{k} p_i^2$. The goal is to estimate this quantity to within an additive error of $\pm \epsilon$.

As mentioned, estimating the second moment of a distribution is a classical problem in statistics and data analysis. However, most of the literature (for example, all of the streaming literature) focuses on multiplicative
$(1+\epsilon)$-approximation of these quantities, rather than additive approximation. In many interesting cases, an $\epsilon$-additive approximation is much more crude (and consequently, much easier to compute) than a
$(1+\epsilon)$-multiplicative one. Indeed, it is not hard to see for example that the collision probability is smaller than
$\max \{p_1, \ldots, p_k\}$, so for any distribution that doesn't have an heavy hitter with probability more than $\epsilon$, returning the value $0$ or $\epsilon$ would trivially satisfy the approximation requirements.

The main two contributions of the authors are as follows:
1. A differentially private estimator for the collision probability in the local DP model. The sample complexity for local $(\alpha, \beta)$-DP estimation to within additive error $\epsilon$ is $O(\log (1/\beta) / \alpha^2 \epsilon^2)$, an improvement by a factor of $1/\epsilon^2$ over previous work [Bravo-Hermsdorff et al., NeurIPS 2022]. Notably the results in Bravo-Hermsdorff et al. are in the pure DP local model, whereas here the results are in the approximate setting.
2. An estimator, not differentially private, for the sequential setting where we do not know $\epsilon$ in advance and need to test whether the collision probability is equal to some value $c_0$ (null hypotheses) or $\epsilon$-far from it in additive terms. The sample complexity, for constant success probability, is $O(\log \log (1 / \epsilon) / \epsilon^2)$, and it is standard and easy to see that this is optimal up to the lower order term.

**Strengths:**

1. The writing is clear and intuitive. It was easy to follow the technical arguments, and I believe they are correct (did not look at some of the full proofs though).
2. Experimentally, the mechanism proposed here seems to outperform the existing approach from Bravo-Hermsdorff et al., and slightly outperform the open source implementations for locally private heavy hitters.

**Weaknesses:**

1. Poor contextualization of the results in comparison to previous work. Why focus on additive approximation? What are precisely the best known results?
2. Limited novelty, both results are simple and do not introduce significant new tools.

**Questions:**

Disclaimer: I have reviewed this paper in the past and one request I had still remains. Please contextualize your paper better against the problem of locally private heavy hitters. It seems that your problem is in some ways easier and perhaps less interesting than heavy hitters. Does a separation between collision probability and heavy hitters follow from existing results in the literature (or from your existing results)? can you show that your problem is provably easier in terms of sample complexity?

---

> ### Author Response · Authors · 2023-11-23
> **Response**
>
> Thank you very much for your careful review!
>
> **Regarding additive error guarantees:** Butucea and Issartel (2021) and Bravo-Hermsdorff et al (2022), the closest previous work to ours, also proved additive error guarantees for privately estimating collision probability. The reviewer is correct that when no element of the domain has probability higher than $\epsilon$ then outputting zero satisfies an $\epsilon$ additive error guarantee. But determining whether this is the case is non-trivial.
>
> **Regarding the difficulty of the problem:** In Appendix K of our latest revision we give a proof that our private collision probability estimator has significantly better sample complexity than the alternate approach suggested by the reviewer (and tested in our experiments) of reducing the problem to private distribution estimation.

---

> > ### Comment · Reviewer_cQaE · 2023-12-04
> > **Thanks for response**
> >
> > Thank you for your response -- I acknowledge having read it.

---

### Official Review · Reviewer_vjJb · 2023-10-21

**Soundness:** 3 good
**Presentation:** 3 good
**Contribution:** 3 good
**Rating:** 6
**Confidence:** 4

**Summary:**

Given samples from a probability distribution $p$ on a universe of size $k$, the collision probability is defined as $\sum_{i=1}^k (p_i)^2$. This paper studies the problems of privately estimating and sequentially testing the collision probability. This paper first gives a locally differentially private algorithm for estimating the collision probability up to additive error $\varepsilon$, provided the number of samples is roughly $O\left(\frac{1}{\varepsilon^2}\right)$. The paper then gives a tester for collision probability that distinguishes whether the input distribution has some collision probability $c$ or is $\varepsilon$-far from $c$, using roughly $O\left(\frac{1}{\varepsilon^2}\log\log\frac{1}{\varepsilon}\right)$ samples.

The LDP mechanism chooses a salt uniformly at random and sends a one-bit hash by applying a hash function to the dot product of the salt and user's sample. The server then partitions the users into a number of groups, compares the collision frequency within each group, corrects the bias, and outputs the median across the groups.

The tester similarly computes the number of pairwise collisions, corrects the bias, and rejects if the resulting estimated collision probability is too large from $c$.

**Strengths:**

- The private collision probability estimator is relatively accessible
- The correctness guarantees seem fairly technical due to possible correlations in the hash function for the estimator or the martingale for the tester
- There are a number of simple experiments that serve as a proof-of-concept to complement the theoretical results of the paper

**Weaknesses:**

- The experiments are on simple uniform and power law distributions rather than specific datasets that would help demonstrate the applications of collisions probability estimation

**Questions:**

- I'm not sure I fully buy the claim that $\varepsilon$ is not necessary as an input to the collisions probability tester, since I think that at some point the algorithm does actually need to output something rather than run on indefinitely, even though the technical statements show a "for-all" guarantee that shows it would eventually reject for all values of $\varepsilon>0$. Perhaps the authors can address this point during the discussion phase.

- As there results that show that collision statistics suffice for uniformity testing, is it possible to extend any of the results in this paper to uniformity testing?

[DGPP19] Ilias Diakonikolas, Themis Gouleakis, John Peebles, Eric Price: Collision-Based Testers are Optimal for Uniformity and Closeness. Chic. J. Theor. Comput. Sci. 2019 (2019)

---

> ### Author Response · Authors · 2023-11-23
> **Response**
>
> Thank you very much for your careful review!
>
> **Regarding experiments on synthetic distributions:** Since collision probability is invariant to any reordering of the domain elements, we may assume without loss of generality that for any distribution $\mathbf{p}$, the probabilities $p_i$ decrease with $i$. Making this assumption clarifies that the only possible difference between real and synthetic distributions is the rate of decay of the $p_i$’s. In our original submission, we explored various decays by testing on both uniform and power law distributions (where $p_i \propto 1/i$). In our latest revision, we added experiments on the exponential distribution (where $p_i \propto \exp(-i)$). See Appendix J of the latest revision.
>
> **Regarding whether $\epsilon$ can be unspecified for sequential testing:** We agree that in practice any testing algorithm must eventually be stopped, for example, when the sampling budget is exhausted. However, our theoretical guarantee ensures that the algorithm will not stop itself when the null hypothesis is true, and will eventually stop itself when the null hypothesis is false, without the user needing to know the value of $\epsilon$. Thus if the algorithm is prematurely stopped for practical reasons, the test will be inconclusive, but it will not produce a wrong answer (with high probability).

---

> > ### Comment · Reviewer_vjJb · 2023-12-04
> >
> > Thank you for the follow-up. I acknowledge having read the response.

---

### Official Review · Reviewer_PG2x · 2023-11-01

**Soundness:** 3 good
**Presentation:** 2 fair
**Contribution:** 2 fair
**Rating:** 5
**Confidence:** 3

**Summary:**

This work considers the problem of estimating or testing collision probabilities given samples from an unknown distribution in two interesting settings: local privacy and sequential testing. For the former problem, the goal is to develop a sample- and communication-efficient mechanism that achieves $(\alpha,\beta)$-privacy in the usual DP sense, with the local constraint ensuring that privacy is maintained even with an untrusted server. When $\alpha,\beta>0$, this work extends prior work of Bravo-Hesdorff, et al. [Neurips'22] by providing a distributed hashing-based approach that achieves sample complexity $O(\log(1/\beta)/\alpha^2\varepsilon^2)$ to attain $\varepsilon$ additive accuracy with constant probability. This is complemented with a lower bound for the case $\beta=0$, where it is shown that the denominator is necessary. The sequential testing problem is as follows: given a value $c_0\in [0,1]$ and error probability $\delta$, query samples from the distribution and either (i) never reject with probability at least $1-\delta$ if the true collision probability is $c_0$, or (ii) if the true collision probability is at least $\varepsilon$-far from $c_0$, reject after $N(\varepsilon,\delta)$ samples with probability $1-\delta$ with minimal $N(\varepsilon,\delta)$. In this case, this work shows that $N(\varepsilon,\delta)\leq O(\log(\log(1/\varepsilon)/\delta)/\varepsilon^2)$ is sufficient, while $\Omega(\log(1/\delta)/\varepsilon^2)$ is necessary. The upper bound is attained by leveraging powerful martingale inequalities to analyze the deviations of an appropriate estimator.

**Strengths:**

This paper is quite well-written and respectful to the reader. The privacy guarantees appear to be novel and the algorithms themselves are quite practical, which is further exhibited through experimental evaluation.

**Weaknesses:**

Some of the comparison to prior work appears at first glance to be misleading (see Weaknesses). While the sequential testing problem for collision probability does not appear to have been explicitly considered in the literature, it seems to follow from existing results.

**Questions:**

Recommendation: While the paper is nicely written, my current understanding is that the technical achievements of this paper are somewhat overstated or unclear at present. As a result, my current inclination is weak reject; however, I could very well be convinced by the authors or other reviewers that my understanding is flawed, and I would be more than happy to revise my score in this case.

Major Comments/Questions

---It may also be worth giving more motivation for why sequential testing (i.e. where the cutoff parameter $\varepsilon$ is not given as an input) is a relevant goal compared to more standard objectives in property testing.

---What specific properties of the hash family do you need? The reason I ask is that communicating a random hash function requires a large number of bits; however, this can be reduced drastically if one can settle for $\ell$-wise independence in the analysis using standard techniques from the pseudorandomness literature (see, e.g. Vadhan's monograph).

---As best I can tell, the lower bound in Theorem 3 is not really comparable to the upper bound in Theorem 2 since one imposes $\beta = 0$ in the former while the upper bound requires $\beta>0$. For instance, when $\beta>0$, it does not seem to be necessary to have any dependence on $\alpha$, since it seems to me that simple approach is to have each agent to uniformly randomize their sample with probability $(1-\beta)$ and then debiasing the resulting collision probability. This would require $\mathsf{poly}(1/\beta)$ samples, so it may also be worth explaining which parameter dependencies are better in practice.

---Unless I'm missing something, the claim that the guarantee of Theorem 2 improves upon the prior result of Bravo-Hemsdorff et al. might be a bit misleading since their result holds for $\beta=0$ whereas Theorem 2 does not. While I'm not an expert, one expects that enforcing $\beta=0$ leads to different or incomparable technical challenges. However, the lower bound in Theorem 3 gives some evidence that the Bravo-Hemsdorff et al. result may be improved.

---I might be missing some subtlety, but can one obtain the sequential testing guarantees using the existing results of Daskalakis and Kawase [ESA'17]? Namely, given a stream of samples $X_1,X_2,\ldots\in [k]$ from some fixed distribution, consider $Z_1,\ldots\in \{0,1\}$ with $Z_t = \mathbf{1}[X_{2t-1}=X_{2t}]$. Then the $Z_t$ are independent coin flips with bias $p$ where $p$ is the collision probability of the distribution. The two cases are then that $p=c_0$ or $\vert p-c_0\vert=\varepsilon>0$. My understanding is that Daskalakis and Kawase [Theorem 2, ESA'17] have shown that a natural sequential testing rule that, given a candidate $c_0$ and the stream $Z_1,Z_2,\ldots$, proceeds by testing whether this bias is close or far to $c_0$ with dyadically decreasing thresholds will properly reject after $\Theta(\log(\log(1/\varepsilon)/\delta)/\varepsilon^2)$ samples of the $Z$ with probability at least $1-\delta$ or never reject with probability at least $1-\delta$ in the good case.

---Relatedly, Theorem 6 is nontrivial compared to the problem of testing (sequentially or not) the bias of a coin from independent samples since one gets strictly more information in this case. One suspects that the extra $\log\log$ term is inherent, but this does not appear to follow from the lower bounds of Daskalakis and Kawase since one gets more information  in this setting (namely, the actual values in $[k]$). Closing this gap would be quite interesting.

Minor Comments

---It may be worth a footnote to explain the difference between ``local'' DP and DP for the non-expert.

---I did not get a chance to check the proofs in the Supplementary Material too carefully, but the proof sketches seemed reasonable.

---While my own interest is more on the theory side, the empirical results in Section 6 appear quite interesting.

---For what it's worth, there is probably a simpler proof of Theorem 7 in the Appendix; since the collision probability is the probability of the event two independent samples are equal, the difference in collision probabilities is at most the total variation between the product distributions $X^{\otimes 2}$ and $Y^{\otimes 2}$. But this is at most twice the total variation between $X$ and $Y$ by a hybrid argument.

---

> ### Author Response · Authors · 2023-11-23
> **Response**
>
> Thank you very much for your careful review!
>
> **Regarding the approximate differential privacy parameter $\beta > 0$ instead of $\beta = 0$:** The DP literature contains many examples of mechanisms that increase $\beta$ from zero to a positive value, while improving the dependence on some other parameter. A classic example is the Gaussian mechanism, which has $\beta > 0$, but also improves performance for multiple queries relative to the Laplace mechanism, which has $\beta = 0$. See https://desfontain.es/privacy/gaussian-noise.html for a readable discussion.
>
> **Regarding a comparison to Daskalakis and Kawase (2017):** Their Algorithm 1 has two significant disadvantages relative to our Algorithm 2 that lead to much higher sample complexities in practice. First, their approach is based on a simple "doubling trick": They repeatedly invoke a non-sequential testing algorithm on subsequences of samples with successively smaller values of the error tolerance $\epsilon$, and stop when the testing algorithm rejects. This is a wasteful use of samples compared to our approach, as stopping cannot occur within a subsequence, and everything learned from previous subsequences is discarded. Second, applying their approach to collision probability testing requires partitioning the $n$ samples into $\frac{n}{2}$ disjoint pairs, so that the observed collisions are independent of each other. By contrast, our approach uses observed collisions among all $\binom{n}{2}$ pairs of samples to estimate collision probability, which significantly complicates the theoretical analysis, but leads to much better performance in practice (our other contribution, an improved mechanism for private collision probability estimation, also uses all $\binom{n}{2}$ pairs of samples to form estimates, while previous work used $\frac{n}{2}$ pairs).
>
> We empirically compared the two algorithms, with the results shown in Appendix J of our latest revision. Our experiments demonstrate that as the null hypothesis approaches the true collision probability, the sample complexity of Daskalakis and Kawase's algorithm is much larger than ours. We will include this discussion and experiments in the final revision of our paper, and we thank the reviewer for raising this issue.

---

### Official Review · Reviewer_aZJ9 · 2023-11-01

**Soundness:** 4 excellent
**Presentation:** 4 excellent
**Contribution:** 4 excellent
**Rating:** 8
**Confidence:** 3

**Summary:**

This paper studies the problem of estimating the collision probability(CP) of a discrete distribution, where CP is the L2 norm of the probability vector. CP estimation is well motivated and has applications to a number of areas in statistics. This work considers the problem in the distributed privacy setting where there are a number of users that have a single sample drawn from a distribution. The aim is to design a mechanism that lets the users communicate their information to a central server, such that the server can determine the CP without learning too much about the sample held by any particular user.

The solution presented in the paper improves upon the state of the art in terms of the minimum number of samples required to ensure the privacy guarantees are met. The proposal is nearly optimal in the sample complexity, as shown by lower bound is also provided in the paper.  The authors implement their algorithm and test it on a dataset, to find that the algorithm performs better in practice as well.

The authors also consider the problem of sequential testing in the standard non-private setting, where one has sample access. Here the task is to test whether the CP is some value c_0 or if it additively epsilon far from c_0. The sequential testing problem also requires that the test be adaptive in terms of epsilon, when the epsilon is unknown. Here they provide an algorithm nearly optimal in epsilon.

**Strengths:**

The presentation is quite clear, and the paper was a joy to read. The key contribution of hashing + salting seems to be independent interest. I appreciate that the authors present the constants in the paper, making it easier to verify the proof.

**Weaknesses:**

I did not quite understand why the problem of private estimation of CP and sequential testing of CP are considered together in a single paper. The methods seem to not overlap much.

**Questions:**

--

---

### Meta-Review · Area_Chair_rUfC · 2023-12-23

**Metareview:**

The authors propose some creative techniques (involving salts and hashes) to estimate the collision probability of a discrete distribution in a locally private manner, and a sequential test for collision probability. These two contributions appear to have little to do with each other (the second being non-private and using different techniques).

While the paper was on the borderline after reading the reviews and responses, after reading the paper and proofs myself, I recommend a rejection. Apart from several missing discussions to related work mentioned by reviewers, I found a bug in the proof of the sequential test, specifically Lemma 5. While the final result might be correct in spirit, the current proof is irreparably and fundamentally incorrect. Below, I propose a fix that the authors can implement in a revision (which may also significantly simplify the proof).

The major bug in the proof of Lemma 5: the invocation of the time-uniform martingale bound of Howard et al. in (12) is incorrect. While it is true that Y_i(p) is bounded, its bound depends on i and increases with i. The stated bound \phi(i,\delta) applies only in the 1-subGaussian case: otherwise there is a price to pay for the subGaussian constant, which in this case even changes with time. So that bound cannot be directly plugged in.

The proof also has several minor typos: Theorem 8.1.1 is attributed several times to Tsybakov, and once to de La Pena and Gine, while only the latter reference is correct. Near the bottom of Pg 21, |Y_k| <= 4m, but perhaps it should be 4k (here is where the bound depends on the index k). After (12), there is a missing bracket for Y_i(p), and it is mentioned that each term is subGaussian, but the subGaussian constant is not mentioned, while here it is particularly important because it increases with time. The definition of \bar U_m might also be missing a normalization by m(m-1).

In my view the above proof technique is not salvageable, though the authors may manage to do this with some effort. A different approach is to directly use the confidence sequences for (possibly degenerate) U-statistics and V-statistics provided in Section 4.2 of "Martingale Methods for Sequential Estimation of Convex Functionals and Divergences, T. Manole and A. Ramdas, IEEE Transactions on Information Theory (2023)". I believe that the lower and upper confidence sequences (for bounded kernels) will be of the same order as the one used from Howard et al, but it might significantly shorten and simplify the proof, since it bypasses the decoupling techniques employed here.

Apart from this, following reviewer suggestions, I recommend that the authors seriously discuss the related work in terms of papers on locally private heavy hitters, and on multiplicative approximation of collision probability, and the possibility of applying these techniques to other related problems (especially since some other cited papers handle several problems at once, not just collision probability).

**Justification For Why Not Higher Score:**

Errors in one of the proofs, and insufficient discussion of related work and techniques in those papers (locally private heavy hitters and multiplicative approximation algorithms for collision probability).

**Justification For Why Not Lower Score:**

N/A

---

### Decision · Program_Chairs · 2024-01-16

Reject